# Establishing a carnivoran of extensive forests on an intensively managed landscape: Habitat and population establishment

**Roger A. Powell**[ID][1]*, **Aaron N. Facka**[1], **Deana L. Clifford**[2,3], **Kevin P. Smith**[ID][1,4], **Sean M. Matthews**[5], **Ed Murphy**[6], **J. Scott Yaeger**[7], **Pete Figura**[4], **Richard Callas**[4]

**1** Department of Applied Ecology, North Carolina State University, Raleigh, North Carolina, United States of America, **2** Wildlife Investigations Laboratory, California Department of Fish & Wildlife, Rancho Cordova, California, United States of America, **3** Department of Medicine and Epidemiology, School of Veterinary Medicine, University of California, Davis, California, United States of America, **4** California Department of Fish and Wildlife, Redding, California, United States of America, **5** Institute for Natural Resources, Oregon State University, Corvallis, Oregon, United States of America, **6** Sierra Pacific Industries, anderson, California, United States of America, **7** United States of America Department of Interior, Fish and Wildlife Service, Yreka, California, United States of America

* rpowell@ncsu.edu

## Abstract

Reintroductions to establish populations usually occur in locations believed to have high-quality habitat to maximize the potential for high population growth rates and long-term population viability. Nonetheless, researchers and managers may have insufficient knowledge of what comprises high-quality habitat or of other requirements for members of a species with low population sizes or how to determine whether these conditions are present at potential reintroduction sites. Locations available for reintroduction may lack optimal habitat but have other characteristics that can benefit a reintroduction. Reintroductions allow rigorous study of reintroduced animals to improve understanding of a species' biology and to inform future management and conservation actions. The fisher, a medium sized carnivoran in the family Mustelidae, is a long-lived (5–8 years) species of concern in western North America due, in part, to the perceived incompatibility of fishers and landscapes commercially managed for timber production. Due to concern about the status of fishers in California, from late 2009 to late 2011 we reintroduced 40 fishers from across northwestern California to the 648 km², privately owned Stirling Management Area that was managed intensively for timber production in the northern Sierra Nevada and southern Cascades of California. The controlled initial conditions facilitated research into other aspects of fisher biology. We monitored reintroduced fishers and their offspring through 2017 to evaluate whether this managed landscape in California, predicted to possess adequate habitat for fishers, could support a new fisher population. Both female and male fishers had high monthly survival (>0.95). On average, 81% of adult females gave birth with a mean litter size of 1.9 ± 0.1 (minimum number of kits ±95% confidence interval). Survival and reproduction rates were constant across years and all vital rates were similar to most extant fisher populations elsewhere in California. By 2013, reproduction was effectively independent of the founding individuals. By 2017, the population was relatively small (n = 119 ± 96–141, 95% credible intervals) but had nearly tripled over the initial number

**Data availability statement:** The data supporting this study are available on Movebank at the following link: https://www.movebank.org/cms/webapp?gwt_fragment=page=studies,path=-study9665331.

**Funding:** The California Department of Fish and Wildlife, U.S. Fish and Wildlife Service, Sierra Pacific Industries, and North Carolina State University are the 4 key cooperators and were responsible for carrying out the research and funding the reintroduction. The funders had no role in study design, data collection and analysis, decision to publish, or preparation of the manuscript.

**Competing interests:** The authors have declared that no competing interests exist.

reintroduced. Stochastic population simulations indicated that the population is unlikely to go extinct within the first 50 years after reintroduction, or 40 years after the completion of field research. Nevertheless, significant habitat changes resulting from wildfire could change those predictions. Thus, sites with landscape conditions similar to our study site and managed similarly for timber production should be considered when planning future fisher reintroductions.

## Introduction

Humans have altered more than 50% of the terrestrial land cover on earth, leading to local decreases and extinctions of native plant and animal populations and to global losses of biodiversity [1–7]. A range of conservation actions, including reintroductions of animal and plant populations, have been used to restore populations [8–11]. Reintroductions are often controversial, are usually expensive, and releases of endangered or threatened species often raise the concern of local landowners, citizens and government agencies. Many factors including small founder populations and poor habitat quality at release locations may prevent establishment of viable, new populations, raising concerns for animal welfare.

To meet the goal of re-establishing viable populations, most reintroductions occur in places with sufficient amounts and quality of habitat to support new, persistent populations. Thus, understanding habitat requirements and obtaining measures of habitat availability and quality are standard steps for evaluating candidate reintroduction sites. Unfortunately, good background information on habitat requirements is often incomplete or totally lacking because target animals often have low population sizes [12–15]. Additionally, a potential reintroduction site might lack good, or even mid-quality, habitat but have other characteristics that compensate for lower habitat quality. Locations that have few predators, have limited access by humans, have supportive landowners, are free from specific pathogens or diseases, or that help to connect disparate populations could be considered as candidate locations even though specific habitat metrics (e.g., vegetation) are objectively at lower values compared to other sites or habitat. Another complicating factor for evaluating a reintroduction where habitat is only adequate is that, for a population that is growing and has not reached carrying capacity ($K$), population growth might not be prevented by mediocre habitat quality, only slowed.

To establish a new population via reintroduction, the population must pass through its "Establishment Phase", the period between the release of founding individuals and the time when population persistence is independent of augmentation from further releases, which we call the "Persistence Phase" (Fig 1) [16–20]. The establishment phase is the period early in reintroductions when populations are most susceptible to extinction because of random environmental or demographic stochasticity, Allee effects, or other factors that are not indicative of the habitat or conditions at the release site. The persistence phase is the period when the population is largely independent of such forces and of augmentation. An incipient population requires sufficient habitat during both phases but, potentially, could fail independent of habitat quality during establishment.

A reintroduction allows extensive control of the founding individuals and, thereby, many characteristics of their new population, allowing researchers to test hypotheses related to habitat quality and habitat choice, to causes of spacing patterns, and to other aspects of basic biology, all of which increase the probabilities of establishing other new populations [18,19,21]. In addition, reintroductions allow control of numbers of founding individuals removed from source populations, thereby allowing tests of the effects of removal on those populations [22]. The consequence is that reintroductions provide conditions to do novel or basic science that

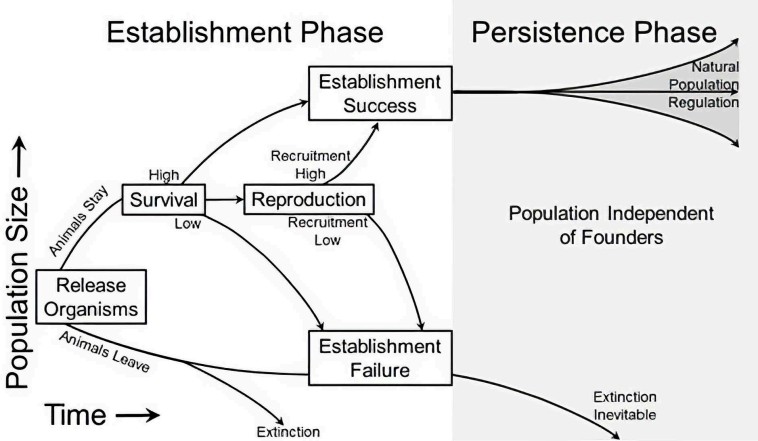

**Fig 1. Phases of a reintroduction with potential outcomes within each of those phases starting from release, animal movements and residency, and short and long-term rates of survival and reproduction.** This model assumes that the reintroduction site is free from immigration that could rescue the population and that successful emigration to another site represents an insufficiency of conditions resulting in establishment failure.

extends beyond the reintroduction itself and that may benefit conservation at broader temporal or spatial scales.

Fishers (*Pekania pennanti*) are medium-sized carnivorans in the family Mustelidae and are native only to North America. Though associated with large stands of late successional, northern forests with complex structure in the understory and continuous canopy prior to, and during, European colonization [23–28] many fisher populations today exist in apparently sub-optimal areas, such as suburbs of major cities, agricultural areas with woodlots, and forested landscapes that are managed for timber production [26,28–33]. In particular, managed timberlands may include large swaths of potentially valuable range, yet how managed forests maintain even low fisher densities has not been investigated extensively.

Across their range, fishers prey primarily on medium-sized mammals (porcupines, *Erethizon dorsatum*; snowshoe hares, *Lepus americanus*; squirrels, *Sciurus* spp.) [26] and birds (grouse, Galinae), that are often found, though not exclusively, in mature northern forests. Such forests appear to facilitate prey capture and minimize risk of predation on fishers themselves [34,35]. Fishers avoid forest patches with sparse overhead canopy (<50%) [36,37] or small diameter trees, which are common after clear-cut logging, avoid including such habitats in their home ranges, and abandon home ranges or parts of home ranges when they are logged [26,29,38,39], though they will use such areas after forests regrow. Mid-successional stands developing from clear-cuts or maintained without letting old trees and snags (standing dead trees) accumulate also lack complex forest structure and coarse woody debris on the forest floor. Female fishers require tree cavities for parturition, which centers around 1 April [26,40,41], after a 10-month delay of implantation and a 2-month active pregnancy [26,42–46]. Logging that removes old trees, especially hardwoods and snags, reduces the availability of cavities.

Fishers have been of particular conservation concern throughout much of their range because of significant population decreases and range contractions through the 19[th] and early 20[th] centuries, due primarily to over-trapping for fur, habitat alterations related to timber harvest, and climate changes at the end of the Little Ice Age [26,27,47–51]. Attempts to recover fisher populations have included at least 40 reintroductions throughout the species' range [31,52]. In their review of reintroductions with known outcomes, however, Lewis et al. [31]

found 89% of fisher reintroductions re-established populations in eastern North America but only 43% did in western North America. Using data from reintroductions, they tested several hypotheses generated by population models and found that only the number of founding fishers and the proximity of the source and original population locations (assumed to represent genetic relatedness) affected reintroduction success. They hypothesized that differences in snow cover, forest succession and characteristics, and predator-prey community structure also contributed to the east-west contrast for re-establishing populations but lacked data to test those hypotheses.

In the mid-2000s, Sierra Pacific Industries (hereafter "Sierra Pacific"), which owned and managed approximately 6600 km² of forest land in California for timber production, collaborated with the USDI Fish and Wildlife Service, the California Department of Fish and Wildlife, and North Carolina State University to reintroduce fishers to the Stirling Management Area (Stirling), a 648 km² portion of Sierra Pacific ownership in the northern Sierra Nevada and southern Cascade Mountains [53]; Facka [18] provided additional background on this reintroduction). In-person evaluations of Stirling and previously published habitat models led the authors of this paper and the funders of the reintroduction to conclude that the amount and quality of habitat available for fishers on Stirling was adequate to support a reintroduction and would allow testing of hypotheses related to re-establishing a fisher population on land managed for timber production.

The goals of the reintroduction were to expand the contemporary distribution of fishers in California, to evaluate the capacity of a landscape managed in accordance with the California Forest Practice Rules [54] to support a reintroduced population of fishers, to gain insight into how changes in forest conditions affect a reintroduced fisher population, and to support research that tested hypotheses related to fisher biology, conservation and management [19,20]. Given that obtaining reliable estimates of vital rates for populations of long-lived species takes as long as 5–7 years [55], the research was planned to last 7 years (later extended to 8 years). We did not know, *a priori*, whether this time period would take the reintroduced population through its Establishment Phase.

A critical aspect of our research on Stirling was to understand how habitat and forest management practices affect fisher survival and reproduction, which led us to a series of hypotheses relating forest management and conditions at Stirling to the fisher population (Table 1). Hypothesis 1) The vast majority of fishers show site fidelity. If released fishers perceived their new environment to be at least minimally suitable, then they would establish home ranges in areas of good habitat on the release site [19,56–59].

Hypothesis 2) Fishers maintain survival and reproductive rates adequate to establish an independent population during its Establishment Phase (not requiring new immigrants). If fishers chose to remain on the release site, then their rates of survival and reproduction should be sufficient to establish and maintain a growing population.

**Table 1.  Hypotheses tested.**

| 1 | Fishers show site fidelity |
|---|---|
| 2 | Fishers maintain survival and reproductive rates adequate to establish an independent population during its Establishment Phase. |
| 3 | Timber management practices by Sierra Pacific affected negatively the survival and reproduction by fishers in the reintroduced population. |
| 4 | Fishers reintroduced onto a landscape managed using even-aged timber production in the northern Sierra Nevada and southern Cascade Mountains show preference for the oldest, most structurally diverse forests on the landscape. |

Hypothesis 3) Timber management practices by Sierra Pacific affected the survival and reproduction by fishers negatively in the reintroduced population.

Hypothesis 4) Fishers reintroduced onto a landscape managed using even-aged timber production show preference for the oldest, most structurally diverse forests on the landscape.

For these 4 hypotheses, the poor record for reintroductions of fishers in western North America dictated that the null hypothesis for each was that fishers would not become established.

Our initial study design focused on how fishers became established, or failed to become established, on our release site and what that process indicated about the habitat quality of the site. Nevertheless, we recognized that stochastic events, and in particular large wildfires, represented a potential threat in both the establishment and persistence phases of our reintroduction. Consequently, we wished to evaluate how the probability of large wildfires could influence persistence of a newly established fisher population. Incorporating habitat use and the rates of survival and reproduction that we documented on Stirling, we used population simulations to explore the potential effects of fire on our reintroduced fisher population. Indeed, since we ended our field research, both the Camp Fire and the Park Fire burned parts of Stirling.

Thus, we expanded the definition of success for our reintroduction to include understanding why the reintroduced population became or failed to become established. With this approach, we maximized the chances of project success even if the reintroduction failed to establish a new population [20].

Previous publications using data from our research have highlighted other important aspects of fisher biology and reintroductions: Overlap of home ranges by conspecifics of opposite sex is forced on females by the much larger males [19]; Stand use by fishers increases with tree squirrel abundance in a stand, which increases with mast production capacity of that stand [60]; Shrubs and non-coniferous trees provide both food and cover for small mammal prey [61]. Female fishers released before 1 January have a significantly higher probability of reproducing than do female fishers released later [21]; A source population for a reintroduction can have up to 20% of its population removed to be reintroduced elsewhere without endangering the source population [37,62].

## Methods

*Study Site*−Our reintroduction site was the Stirling Management Area, located in portions of Plumas, Butte, and Tehama counties in northern California, USA (Fig 2), where the southern Cascade Mountains meet the northern Sierra Nevada. Elevations ranged from 425 to over 2400 m. The climate was temperate with the majority (>85%) of precipitation coming in late fall and winter as snow fall and rain [63]. Vegetation was typified by Sierra Nevada mixed conifer forest dominated by ponderosa pine (*Pinus ponderosa*), sugar pine (*Pinus lambertiana*), incense cedar (*Calocedrus decurrens*), white fir (*Abies concolor*), Douglas fir (*Pseudotsuga menziesii*), and California black oak (*Quercus kelloggii*). Tanoak (*Notholithocarpus densiflorus*) and canyon live oak (*Quercus chrysolepis*) also formed dense stands in some locations [64,65]. Stirling was bordered to the north and east by the Lassen and Plumas National Forests, managed for multiples uses, and to the north by Collins Pine Company lands, managed for timber production most often using uneven aged management.

Stirling had a history of diverse forest management. From the early 1900s through 1999, the property was managed primarily with single tree selection. Additionally, small areas burned by wildfires were logged to salvage timber and replanted. In 1999, even-aged management became the primary silvicultural strategy and nearly all merchantable trees were removed at each harvest by clear-cutting. Starting in 2004 as the company policy, and formally

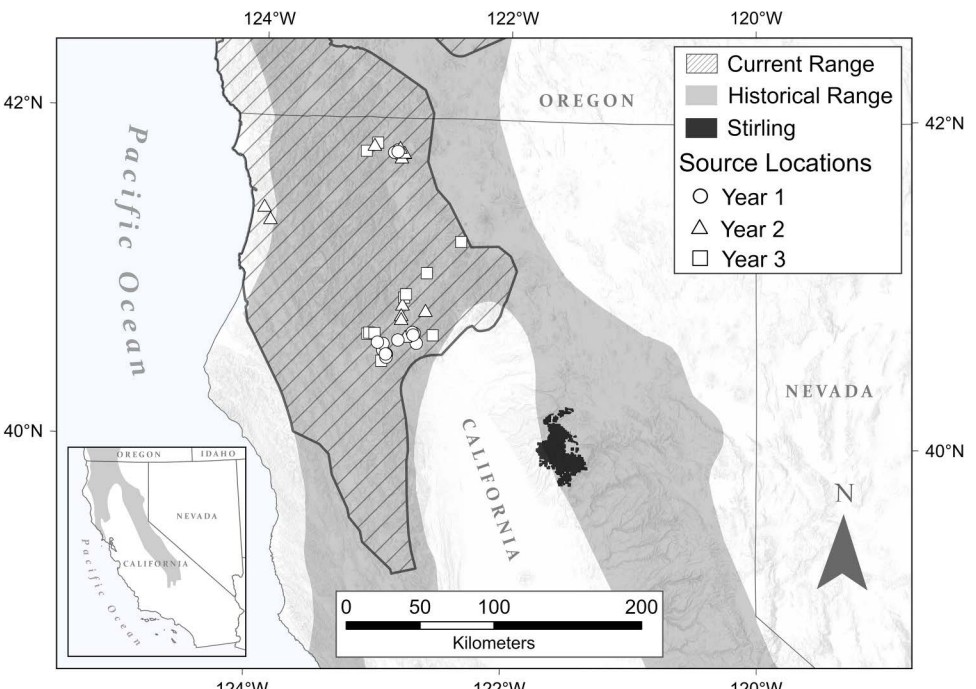

**Fig 2. Locations, by years, where we captured fishers for release onto the Stirling Management Area of Sierra Pacific Industries in northern California overlain with the estimated historical [ 127] and pre-reintroduction distributions of fishers [31].**

in 2008 as part of a Candidate Conservation Agreement with Assurances with the USDI Fish and Wildlife Service, a patch totaling 2% of each stand designated for clear-cutting was left uncut, with the goal of retaining snags, coarse woody debris in the understory, live hardwood trees and other diversity within the stand, the rest of which was regenerated with planted conifer seedlings. California Forest Practice Rules required, with few exceptions, that no more than 8 hectares (20 acres) were harvested in any single clear-cut and that adjacent stands must be retained for 5 years [54]. Further, Sierra Pacific's company policy required that stands adjacent to clear-cuts be retained for 10 years and required retention of structures, such as snags and live hardwoods. Stands were thinned pre-commercially at 10 years post-harvest and again at approximately 40 years. Approximately 5–15% of a watershed was harvested each decade over a planned 80-year rotation. When the fisher reintroduction began in late 2009, approximately 43% of Stirling was in early successional forest (small trees and early seral stands) and 29% was in late successional forest (medium tree with open canopy and large trees with closed canopy; Fig 3).

*Live-capture and moving fishers*–Capture of fishers for reintroduction on Stirling began in late November 2009. We captured fishers from well separated locations in northwestern California to minimize the impact to any 1 source area and to maximize genetic diversity of the founding population (Fig 2) [53]. We transported all captured fishers to a central processing area and evaluated them for release onto Stirling.

We immobilized fishers with Tiletamine HCL and Zolazepam HCL (Telazol®, Fort Dodge Animal Health, Fort Dodge, Iowa, USA) at dosage of 7 mg/kg to collect data on sex, age, reproductive status, general condition, disease exposure, weight, and morphology and to fit transmitters. We vaccinated all fishers against canine distemper virus with Purevax® (Merial Limited, Duluth, Georgia, USA) and against rabies virus with Imrab® 3 (Merial Limited,

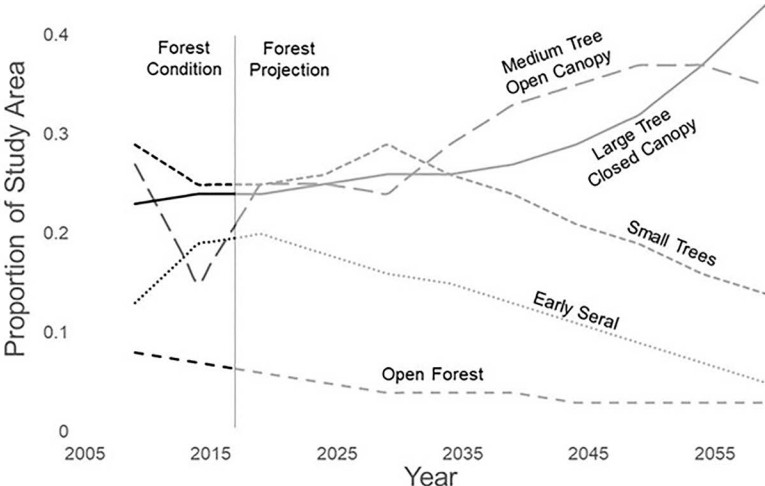

**Fig 3. Proportion of the Stirling Study Area in 5 forest classes.** Data through 2017 (black lines) represent the actual forest conditions. Data from 2018 (grey lines) through 2050 are projections supplied by Sierra Pacific Industries.

Duluth, Georgia, USA) and we treated them for endo- and ecto-parasites with Ivermectin and Frontline® (Merial Limited, Duluth, Georgia, USA). We held fishers in captivity for 3–14 days while we tested them for previous exposure to canine distemper virus and canine parvovirus [66]. We returned 1 fisher that tested positive for distemper to his original capture location and held 1 female in captivity until she no longer shed canine parvovirus. The anal glands of several fishers from 1 coastal trapping site were infected with an unknown, undescribed trematode, leading us to reject these fishers for reintroduction and to stop trapping at that site.

For founders on Stirling, we selected female fishers estimated to be approaching their 2nd or 3rd birthdays because they had highest reproductive value ($v_x$) [21,67], thereby having the greatest potential to contribute offspring to the new population over several years. We selected male fishers estimated to be at least 3-years old because large, adult males appear to be more successful breeders than young males [28,31]. We removed 1 upper pre-molar from each fisher to estimate age by counting cementum annuli [68,69] but we estimated ages of fishers in hand using sagittal crest development, tooth wear, and teat size (for females). Our goal was to reintroduce a total of 40 fishers with a sex ratio of 60:40 [53].

To test our 4 hypotheses, tracking fishers' movements was critical and allowed us to document where fishers settled, the timing and rates of mortality, where female fishers denned and produced kits, and to calculate all fishers' utilization distributions. In the initial year of the reintroduction, we were concerned that an external collar might hinder females' abilities to enter tight cavities used as reproductive dens. Consequently, we implanted female fishers surgically with Telonics very high frequency (VHF) transmitters (IMP-310, Mesa, Arizona, USA). In subsequent years, based on our own experience and experiences of other researchers, we outfitted female fishers with small VHF collars (Holohil MI-2i, Carp, Ontario, Canada; Telonics MOD 125) to reduce the stress of surgery. At that time, Global Positioning System (GPS) collars were too heavy or too short-lived to meet our needs. For male fishers early in the reintroduction, we feared that long distance movements could make documenting mortality using VHF transmitters impossible. Consequently, we outfitted adult males with Platform Terminal Transmitter collars (PTT [Argos]; Kiwisat 202 or 303, Sirtrack, Havelock North, New Zealand). We programmed the PTT transmitters to collect at least 1 location estimate per day. PTT collars were active during different time blocks to allow inference about male

locations and movements throughout a 24 hour cycle. To test Powell's [70] model of energy expenditure by fishers, we outfitted 11 fishers with global positioning systems (GPS) collars (Minitrack, Lotek Wireless, Newmarket, Ontario, Canada) during the autumns of 2012–2017 and set those collars to collect a location at <15 minute intervals over 10 days.

At least 1 of our biologists (ANF, JSY, RAP, RLC) and our veterinarian (DLC) evaluated each fisher we considered as a potential founder. We transported all fishers chosen as founders to Stirling within 2 days of making our final decisions. We released fishers onto Stirling in groups of 1–5 individuals, releasing the first 15 fishers (9 F, 6 M) in December 2009 and January 2010 (Fig 4). We released an additional 13 fishers (7 F, 6 M) in November 2010 through February 2011 and the final 12 (8F, 4 M) in November and December 2011 [19,21]. We released fishers throughout Stirling but prioritized central locations that we evaluated to have good habitat (Fig 4) to encourage a widely distributed population and to reduce potential intra-sex conflict [19,71]. We "hard" released fishers without acclimation to their new environments because no data suggested that "soft" releases improved survival of fishers or influenced population establishment [31,52].

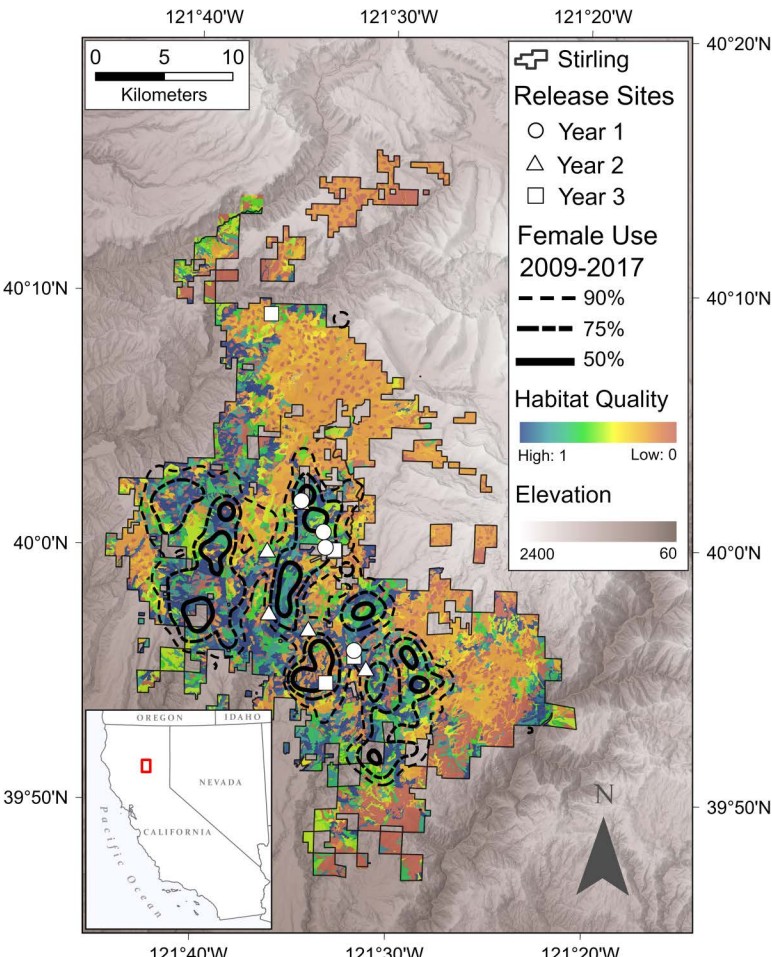

**Fig 4. The 90%, 75% and 50% density contour lines (black lines) for all locations of female fishers combined from 2009 through 2017 overlaid on the mean of the habitat quality estimates using Thomasama's habitat model for 2010 through 2017.** Shaded regions outside of the Stirling boundary represent topographic relief and white shapes are the release locations of all translocated fishers.

Beginning in 2010, we live-trapped fishers annually in autumn to re-collar individuals, to estimate population size, to monitor health, and to mark offspring born on Stirling. These activities provided data independent from telemetry for testing our hypotheses. We trapped over the full extent of Stirling and on adjacent lands within 2 km [37]. In each year, we used between 90 and 110 live-traps that we spaced roughly 1.6 km apart (range 0.5 to 4 km). Spacing and the numbers of traps were similar among years. Live-captures occurred over 14 days in 2011 and 28 days in the following years. We baited each trap with a chicken drumstick and placed a scent lure (Gusto, Minnesota Trapline Products, Pennock, Minnesota, USA) at trap locations. We checked all traps daily and replaced bait and scent as needed. We re-vaccinated founders for the first 2 years after their release, if we caught them, for canine distemper and rabies but we vaccinated fishers born on Stirling only through 2011. We handled fishers captured on Stirling with the same methods used for fishers considered for being founders. Capture and handling methods were approved by the Institutional Animal Care and Use Committee from North Carolina State University (09–007-O and subsequent) and were consistent with the guidelines of the American Society of Mammalogists [72,73].

*Tracking Fishers*–We attempted to locate all fishers with transmitters daily by triangulation [e.g., 74], calculating location estimates using program Location of a Signal (LOAS 4.0, Ecological Software Solutions LLC, Hegymagas, Hungary). We followed telemetry signals to "walk-in" on some fishers until we saw them or could identify occupied structures (*e.g.*, tree, snag, log, rock crevice). We used location estimates from PTT data processed through the Argos system with Kalman estimation and filtering (Collecte Localisation Satellites; Ramonville-Saint-Agne, France). In the first 3 years of the reintroduction, we used fixed-wing and helicopter surveys at least 4 times per year to survey >20 km away from Stirling and document animals that settled, or died, outside of the study area. Thereafter, We used aerial surveys 2 times per year to locate fishers [19,46].

Both VHF and PTT transmitters were equipped with mortality sensors. We walked to transmitters broadcasting mortality signals as soon as possible to confirm a fisher's mortality and to increase our chances of determining its cause of death [74].

To estimate telemetry error for triangulations of VHF data, we compared location estimates from triangulations to "walk-in" locations obtained on the same day (usually within the same hour) for dens and rest sites. This yielded a mean error of 102 ± 132 m (±SD). We assessed true error rates for Argos locations of each error class by comparing satellite locations to known locations of males held in captivity, of collars that had been dropped (the day they are dropped was known from activity data), or of dead fishers. The mean error estimated across all error classes was 767 ± 1241 m. Our calculated mean error for locations in each error class are consistent with expected error predicted by the Argos service [75]. For error class 3 (locations with the most confidence), 91% of locations were within 350 m of the true location. Mean error rates were incorporated in our analyses of animal use of habitats to account for our uncertainty of final locations (Hypothesis 3).

To document site fidelity (Hypothesis 1), we used live-trapping data, VHF telemetry location estimates with error radii < 100 m, Argos location estimates with error radii < 1500 m, den photographs, and den telemetry. We then evaluated what proportion of an animals home range overlapped with the Stirling boundary by at least 50%

*Home Ranges (Hypothesis 1)* – We defined site fidelity as having established a home range [76]. To estimate home ranges and population size, we established a fisher's year to be 1 October through 30 September of the following calendar year. Juvenile fishers disperse in autumn, meaning that around 1 October all fishers are living independently [24,41].

We defined an animal's home range to be that part of the landscape in which it lived and for which it updated its cognitive map of the landscape [77]. We assume that the 95% utilization

distributions for fishers' use of their environments provide acceptable, though flawed, estimates of home ranges [77,78]. We calculated utilization distributions for fishers having ≥ 30 location estimates per year [79–82]. We recorded ≥ 100 estimated locations per year for most fishers.

We estimated utilization distributions using a fixed kernel smoothing program that weighted each location estimate by autocorrelation [83]. We used Silverman's [84] kernel "k2", which is a bell-shaped kernel with finite bounds and is leptokurtic, thereby resembling the distribution of telemetry error for experienced researchers [76,85]. To choose "h", which represented the width of the kernels, we did not use software that selects an "h" that minimizes internal statistical error (such as least squares cross validation) because biological considerations are more important [76,80]. To gain a perspective on values of "h" that other researchers have found satisfactory, we performed a literature search using the terms "((*Pekania* or fisher) and (home and range) or territor*)", yielding 65 publications, only 1 of which, Facka and Powell [19], stated the value of "h" used and why. Consequently, we used aspects of fisher biology and telemetry error to choose "h" but, unfortunately, different aspects of fisher biology suggested different values for "h". The mean radius for 95% of our fishers' movements was roughly 2.5 km, suggesting that about $^2/_3$ of their movements would be within a radius of 1000 m. Maximum daily movement distance was roughly 1500 m, while the mean of daily movements was roughly 750 m. All of these distances exceed our mean VHF and Argos telemetry error, which sets the minimum value for "h". With only 1 location per day per fisher, using mean telemetry error for "h" led to disjunct utilization distributions. Consequently, we calculated utilization distributions using 3 values of "h": 750, 1000 and 1500 m.

*Reproduction (Hypotheses 2, 3)* -- We located females with greater frequency from mid-March through May to locate reproductive dens and to document parturition as detailed by Facka et al. and Smith et al. [21,46]. Consistent with Green et al. [45] and Smith et al. [46], we address birthing dens as "natal dens" and subsequent dens as "maternal dens". We confirmed reproduction by placing 1–6 remotely-triggered cameras (PC800 and PC900 Hyperfire Professional, Reconyx, Inc., Holmen Wisconsin, USA) within 6 m of and facing the base of a suspected den tree to photograph a female and her kits. Photographic and telemetric evidence indicating repeated use of a tree suggested that a female was attending kits. Photographs of females with kits confirmed parturition and documented minimum litter sizes. We collected additional data on birth rate each autumn by examining the teats of captured females for signs of lactation [86].

We examined reproduction as a composite variable (reproductive output) that included whether a female gave birth and the number of kits that we observed. We classified adult females for each denning season as: 1) did not give birth, 2) gave birth and had 1 kit and 3) gave birth and had at least 2 kits. Females that were 1-year-old and, consequently, not old enough to give birth in April, were not included in these analyses. We hypothesized that a female's reintroduction status might affect reproductive rates [21], as might her weight in fall or winter before the denning season and as might her age on 1 April. We analyzed denning rate as a multinomial response using polytomous (multinomial) logistic regression (proc logistic, Statistical Application Software [SAS], Cary, North Carolina, USA). We assessed females annually, even if they were tracked for multiple years, because our main goal was to investigate broad-scale changes in reproduction through time. To evaluate how the models that we tested compared to simple random variation, we included a normally distributed, randomly generated covariate to include within our analyses (random). We converted the beta estimates for the direction and magnitude of effects to odds-ratios using an exponential transformation ($e^\beta$).

To rank hypotheses related to reproduction, we used a generalized linear mixed model using a Poisson distribution and including individual females as random effects to generate

values for AICc. We used the maximum litter size recorded for each female fisher as the dependent variable, assigning a value of 0 to females who did not reproduce. We included models for reintroduced females producing fewer kits than Stirling-born females, female age, proportion of a female's utilization distribution that had timber harvested within the preceding 10 years, habitat quality at a female's locations during the preceding year, the proportion of hardwoods found within a female's utilization distribution (representing availability of cavities for dens) [46], and, finally, year of the study as a continuous variable.

*Mortality and survival (Hypotheses 2, 3)* – For estimating survival and reproductive rates and for projecting population sizes, we defined a fisher year as 1 April through 31 March of the following year. To ascertain potential sources of mortality, we took notes and photographed mortality sites. Fisher carcasses with sufficient remains and with little to moderate autolysis were necropsied by wildlife pathologists at the California Animal Health and Food Safety Lab at the University of California Davis, the Integral Ecology Research Center (Davis, California) or the Wildlife Investigations Lab of the California Department of Fish and Wildlife. Autopsies included tests for anticoagulant rodenticides [66,87]. For fisher carcasses with evidence of predation, we measured puncture wounds and collected matted fur with possible predator saliva, used polyester swabs to swab apparent puncture wounds, bones and the transmitter collar (if no other samples were available) for molecular forensics conducted by the Integral Ecology Research Center to determine the species of predators that contacted the carcass [88]. In cases with scant remains, DNA could have been left by predators or scavengers.

We analyzed monthly and annual survival for all fishers outfitted with transmitters using "known fates" analyses within program MARK [89]. Fishers not known to be either alive or dead within any month were censored. We documented survival of kits from photos of reproductive dens. We estimated kit (= birth to independence, approximately 6 months) survival from birth until capture in the first autumn after birth indirectly in two ways. First, when females with kits died before their kits were 6 months old, we assumed that the kits also died. Therefore, an initial estimate of 6-month survival of kits is the monthly survival rate of reproductive females raised to the 6th power. Second, after live-trapping each autumn, we estimated kit survival by dividing the total number of young-of-the-year captured by the total number of kits photographed at reproductive dens. For this second estimate, if we captured more kits in any year than had been documented we set survival for that year at 1.00. We multiplied the average of these 2 estimates of 6-month survival by the known fates estimate for the second 6 months of a kit's first year to estimate first year survival.

We used Akaike's Information Criterion corrected for small sample size (AICc) to rank the abilities of hypotheses to describe the pattern of mortalities and survival that we documented (Table 2) [90]. We modelled survival first as constant through time, with affects due only to sex and to 3 sex×age classes: juveniles (<1 year old); yearlings (1 age <2); and adults (≥ 2 years old). Additionally, we modelled survival as changing by month×year, treating year first as a class variable, then as a continuous variable, and finally with male and female survival potentially differing across study years, again with year as a class variable. We also incorporated a reintroduction effect to evaluate if reintroduced fishers survived differently than fishers born on Stirling.

To limit the number of covariates related to habitat, we calculated habitat quality within the 50% isopleth of the utilization distribution for each individual fisher, calculated the habitat quality for each location of each fisher, and calculated the proportion of each fisher's utilization distribution that had had timber harvest within the preceding 10 years. We ranked models of survival that incorporated habitat quality to a null model with constant survival.

In a previous publication [37], we estimated population size using spatial mark-recapture. To supplement those analyses, we used fisher live-capture data and telemetry data for

**Table 2. Comparison of 12 models with variables hypothesized to affect survival of fishers on the Stirling Management Area of Sierra Pacific Industries in the northern Sierra Nevada and the southern Cascade mounts of California 20109–2017.** Only models with AICC ≤ 4.0 are shown. Analysis was a known fates analysis in program MARK based on monthly fates of reintroduced fishers and their offspring. "Habitat" refers to Thomasma's index of habitat quality applied within fishers' utilization distributions calculated for each year using data supplied by Sierra Pacific Industries. "Founding Fisher" refers to whether a fisher was a founding fisher or was born on Stirling. "Habitat GNN" refers to Thomasmas's index calculated using the Gradient Nearest Neighbor data set, which was static for the study period. "Proportion Early Seral" refers to the proportion of a fisher's utilization distribution that was recently logged and in early seral stage. Year was a continuous variable.

| Model | $AIC_C$ | $AIC_C^\Delta$ | Likelihood | $w$ | K |
|---|---|---|---|---|---|
| Age | 344.12 | 0 | 1.00 | 0.28 | 2 |
| Founding Fisher | 345.86 | 1.74 | 0.42 | 0.12 | 2 |
| Age + Habitat | 345.96 | 1.84 | 0.40 | 0.11 | 3 |
| Null | 346.08 | 1.96 | 0.38 | 0.10 | 1 |
| Habitat GNN | 346.98 | 2.86 | 0.24 | 0.07 | 2 |
| Sex + Founding Fisher | 347.34 | 3.22 | 0.20 | 0.06 | 4 |
| Age + Founding Fisher | 347.44 | 3.32 | 0.19 | 0.04 | 4 |
| Proportion Early Seral | 347.68 | 3.36 | 0.17 | 0.04 | 2 |
| Age + Sex | 347.55 | 3.43 | 0.17 | 0.04 | 4 |
| Habitat | 348.00 | 3.88 | 0.14 | 0.04 | 2 |
| Sex | 348.03 | 3.92 | 0.14 | 0.04 | 2 |
| Year | 348.04 | 3.92 | 0.14 | 0.04 | 2 |

2011–2017 to calculate for each of our autumn live-capture sessions the minimum number of fishers alive (Fig 5), an index known to be biased low for population size but unbiased for population dynamics [91]. We then used a 4-stage Leslie matrix model (Fig 6) and stochastic simulations of the female portion of the reintroduced fisher population on Stirling to explore the potential effects of juvenile survival on population viability, because our estimates of juvenile survival have the largest potential error. We compared population projections using 2 low estimates of mean juvenile survival (0.2 and 0.3) based on values in the literature [92,93], a medium value of 0.5, and a high value of 0.6, which is less than the estimate calculated from our data. We projected the population for 50 years starting with the reintroduction of fishers (Fig 6) [21,93].

Finally, we iteratively increased and decreased by 10% the values of vital rates across all life-stages and the values of carrying capacity, *K*, denning rate and dispersal rate to estimate the elasticities. Measuring such elasticity quantifies the potential for each variable to affect population growth or extinction.

*Habitat availability and selection (Hypothesis 4)*– We evaluated habitat selection by fishers using forest inventory data provided by Sierra Pacific. These data were based on 39,871 variable-radius plots that were spaced roughly every 80.5 m (North-South) by 201.2 m (East-West) or approximately every 1.6 ha [94]. Each plot was sampled every 10 years staggered by year and estimates of tree growth were used to model tree size, stand volume, canopy cover, and numbers of trees by species annually. Sierra Pacific categorized forest cover as 1) early seral (recently clearcut), 2) small tree (quadratic mean diameter QMD < 15 cm), 3) open forest (canopy > 40%, QMD>= 15 cm), 4) medium tree forest (high canopy>50%, QMD>=28 cm), and 5) large tree forest (canopy >60%, QMD >= 28 cm; Fig 3).

We estimated habitat quality on Stirling using the version of Allen's [25] fisher habitat suitability model modified by Thomasma et al. [95]. The Thomasma index has been tested independently at other study sites and is useful for describing fisher habitat selection [26,36,95,96]. This model quantifies fisher habitat quality based on 4 vegetative metrics: 1) percent tree canopy closure, 2) mean diameter at breast height (DBH) of overstory trees, 3) tree canopy

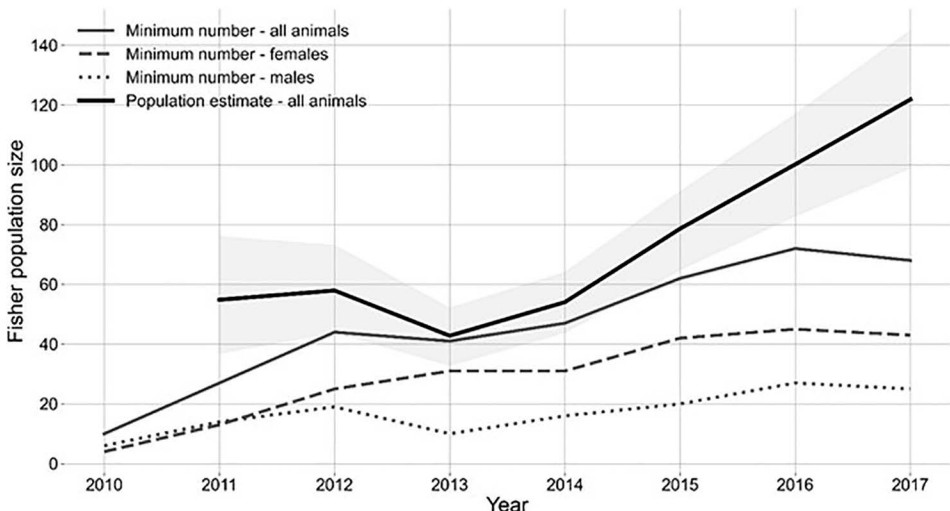

**Fig 5. The minimum numbers of fishers known to be alive on the Stirling Management Area of Sierra Pacific Industries in northern Sierra Nevada and southern Cascade Mountains of California, 2010-2017: total population (solid gray line), females only (gray long dashed line), males only (short, dashed line).** For comparison, the total population estimates from spatial capture-recapture estimator for 2011-2017 [37] (thick black line; shading shows 95% credible intervals).

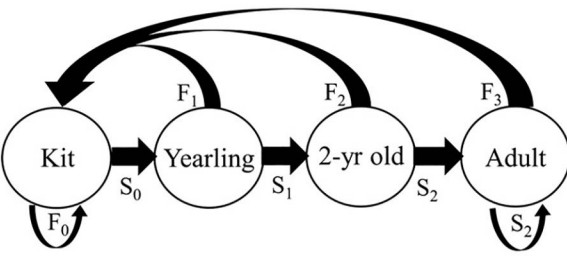

**Fig 6. Life-stage model of the fisher life cycle used to construct projection matrices to simulate reintroductions.** Each circle represents the discrete life-stage of a female fisher with the kit life-stage consisting of all females ages 0 to <1 year old, the yearling life-stage is females 1 to <2, and the adult life-stage is all females ages 2 and older. The arrows connecting the circles represents the transition values between the life-stages: $S_0$ = probability of surviving from birth until 1st birthday, $S_1$ = probability of surviving between 1st and 2nd birthday, $S_2$ = probability of surviving in to successive birthdays after the 2nd, $F_0$ = probability of a newborn giving birth on her 1st birthday (always 0), $F_1$ = the probability of a yearling female denning and giving birth on her 2nd birthday, $F_2$ = the probability of a 2-yr old female denning and giving birth on her 3rd birthday, and so forth.

diversity (*i.e.*, number of canopy layers, including a ground layer if it exists), and 4) percent of overstory trees that are not coniferous (hereafter hardwoods). These metrics relate directly to fishers' requirements for den sites and prey availability [28]. Proximity among habitat patches and spatial configuration (e.g., fragmentation and interspersion) do not contribute to habitat quality in Thomasma's model. We calculated the Thomasma index for Stirling for each year from 2010 through 2017 using Sierra Pacific stand data (Table 3).

Some fisher locations were not on Stirling and, therefore, we used the Gradient Nearest Neighbor data set (https://lemma.forestry.oregonstate.edu/about) to calculate Thomasma's index beyond the boundaries of Stirling. Habitat quality on Stirling changed yearly but the Gradient Nearest Neighbor dataset was static. We estimated the means and standard

**Table 3.  Mean and maximum habitat quality (±SD), quantified using Thomasma's index (0=not habitat for fishers; 1= best habitat for fishers), for forest stands of different stages. All stages had at least one stand with habitat quality = 0.**

| Forest Stage | Mean Habitat Quality | Maximum Habitat Quality |
|---|---|---|
| Early Seral | 0.001 ± 0.007 | 0.06 |
| Small Tree Forest | 0.20 ± 0.20 | 0.84 |
| Open Forest | 0.02 ± 0.05 | 0.27 |
| Medium Tree Forest | 0.30 ± 0.25 | 0.89 |
| Large Tree Forest | 0.27 ± 0.25 | 0.96 |

deviations of habitat quality from the single Gradient Nearest Neighbor database for each contour line that bounded 50, 75, and 90% of the total volume of the raster that was quantified with each fisher's utilization distribution. We used the Thomasma index value each year to estimate mean and standard deviation of isopleths of the utilization distributions.

We evaluated habitat selection of fishers on Stirling at 2 scales: the landscape (the entirety of Stirling) and within utilization distributions (Johnson's [97] 2nd and 3rd orders of selection). First, we tested the hypothesis that the fisher population selected habitats quantified as being of high quality. Using proc genmod of SAS with a gamma distribution, we compared habitat quality at fishers' telemetry locations to the habitat values at an equal number of locations randomly placed across Stirling. We quantified habitat use vs habitat available by using the modified Ivlev resource selection function, which varies from −1–1:

$$Modified\ Ivlev = \frac{2(used - available)}{(1 + used + available)}$$

where "used" represents the locations where we observed fishers and "available" represents the random locations.

Second, we tested whether individual fishers selected areas quantified to be of high quality within their utilization distributions. We again used the modified Ivlev resource selection function but, in this analysis, we used the mean habitat value for all locations for a single fisher as the "used" term and the mean habitat value found across the 95% utilization distribution as the "available" term. Thus, each fisher in each year had a resource selection value and we used these as independent replicates to evaluate differences in resource selection across sexes and through time.

Lastly, we tested whether fishers' utilization distributions encompassed disproportionately areas that had been clearcut within the preceding 10 years. We calculated all areas that had been clearcut within the 10 years prior to and including the year for which we estimated the utilization distribution of individual fishers and divided this area by the area of the 95% isopleth boundary of the utilization distribution. We tested for differences between the sexes and through time (using years of our study as a continuous variable).

*Population Viability Simulations*– We also used a 4-stage Leslie matrix model to explore the effects on population viability from changing forest conditions projected by Sierra Pacific. For these simulations, we set juvenile survival at 0.6 and we simulated the release of fishers over 3 years to match our actual releases, reducing the mean denning rate for females in year-1 and year-3 as documented by Facka et al. [21]. We assumed that numbers or qualities of males did not limit reproduction.

Thus, we conducted simulations of 2 scenarios

1)  A "Baseline" scenario represented a reintroduced fisher population living on a landscape similar to Stirling in 2010 that experienced no changes of forest conditions nor of habitat

quality for 50 years. Vital rates (reproduction and survival), carrying capacity (*K*) and dispersal were held constant. For these scenarios, we started with the following vital rates (see Results): Survival to age 1 = 0.6; Survival ages 1–2 = 0.81; Annual Survival ages > 2 = 0.81; Birth rate age 2 = 1; Birth rate ages >2 = 1; Denning rate = 0.85.

Our proposal for our real fisher reintroduction did not include potential effects of forest fires on the new fisher population. Since the initiation of the reintroduction, however, many wildfires have burned substantial areas in California, including the Camp Fire in 2018 and the Park Fire in 2024, both of which extended into Stirling. Consequently, we varied the Baseline scenario by simulating the effects of a wildfire in year 10. In subsequent years, the simulated area burned succeeds through seral stages through year 50. Vital rates, *K* and dispersal varied in response to forest changes following the fire. In separate simulations, the fire covered 10% to 60% of the forest in 10% increments. For these scenarios, we started with the vital rates reported in our Results section.

2) A timber "Harvest" scenario represented a reintroduced fisher population living on a landscape with forest cover changing as projected by Sierra Pacific on Stirling, leading to changes in habitat quality with concomitant changes in vital rates, *K,* and dispersal. In the projections, forest cover was categorized as early seral, small tree forest, open forest, medium tree forest, and large tree forest. Forest succession and forest management move forest stands progressively through these 5 stages. In early years, Stirling was roughly 15% early seral, 10% open forest, and 25% each small, medium and large trees (Fig 3). During the first 10 years of simulations, early seral increased to about 20%, open forest decreases to about 5% with the other stages remain mostly unchanged. From year 20 on, early seral, small trees and open forest gradually decrease while medium trees increase to about 35% and large trees to about 45%.

As with the Baseline scenario, we simulated the effects of a wildfire of various extents in year 10.

To calculate *K*, we first placed a grid with 100x100 m cells across Stirling and calculated the habitat quality (using Thomasma's index) for each cell in each year, 2010–2017, using annual stand projections supplied by Sierra Pacific. We summed the cell-specific habitat quality values for each year to obtain a total for habitat quality on Stirling for each year. We then placed the outline of the 95% utilization distribution for each female fisher for each year on the habitat quality map for that year and summed the habitat quality values within the outline, giving us a total for the habitat quality for each fisher for each year. Finally, we calculated the mean total for habitat quality for all female fishers for each year and divided that number into the total for Stirling habitat quality annually, yielding an estimate of the number of female home ranges that Stirling could support each year. We truncated that number to the nearest whole female fisher, yielding estimates for *K* that ranged from slightly over 40 to slightly under 50 female fishers, depending on the year. This approach is similar to the approach used by Mitchell and Powell [98,99] to accurately predict optimal home ranges and to predict carrying capacity for black bears (*Ursus americanus*).

For our "Baseline" scenario, we set *K* = 45. For our "Harvest" scenario, K gradually grew from 45 to 57, calculated as above. For each year of our simulations, we adjusted survival in a logistic manner based on the size of the female fisher population relative to *K* for that year. We did not reduce reproduction because Bulmer's [100,101] extensive analyses of fishers' responses to snowshoe hare population cycles showed that fisher populations responded by changing juvenile and adult survival but not reproduction.

Because Stirling was embedded within a forested landscape, we allowed young-of-the-year fishers to disperse from Stirling. As a simulated population for Stirling approached *K*, young

of the year dispersed to a buffer zone around Stirling in numbers that we calculated using a logistic function. When a simulated population was below $K$, young fishers dispersed into Stirling from the buffer.

We generated random normal deviations from mean values of vital rates based on the documented standard deviations, limiting survival values to vary between 0 and 1. At each time step, we truncated population size to its integer value, *i.e.*, to whole numbers of fishers. For each scenario we replicated 50 years of population change 1000 times. We present mean values (±SD) of the mean female population size across all runs for each year. If a simulated population fell below 5 females, we considered the population to have gone extinct and ended the simulation.

For all simulations, we calculated the percent of replicates that went extinct and the year of extinction. We consider the extinction rate for simulated populations to be an index of the probability of extinction for real fisher populations. Thus, if the extinction index goes up or down as simulated conditions changed, we expect that the extinction probability for a real population would go up or down with such changes but not necessarily matching the extinction index.

## Results

From 2009 to 2011, a total of 40 (24F, 16M) founding fishers were reintroduced to Stirling from across the fisher range in northern California (Fig 2). The average age of fishers at time of release was 2.8 years for females (range 0.6–5.7) and 3.6 years for males (range 0.7–6.8). We released 1 juvenile female and 1 juvenile male, both of which were estimated to be adults at the time of examination but were later confirmed juveniles by cementum annuli. All other fishers (n=38) were >1 years old, capable of producing kits and breeding during spring following release.

Between 2010 and 2017, we captured 133 (77F, 58M) individual fishers born on or near Stirling and, thus, collected data on 173 fishers. Our annual capture success ranged from 1.2% (fishers captured/100 trap days) to 3.7%, with lowest capture success in the early years of the project and the highest in the last year. Our first evidence of reproduction on site was from remote camera photos in autumn 2010 of 1 juvenile male individually recognizable by his chest blaze as not having been a founder. Subsequent to the initial releases, the population remained biased towards females (Fig 5; Green et al. 2022).

At the conclusion of 2017, we had estimated or documented 33,658 fisher locations with acceptable error (GPS 44%, triangulation 33%, Argos 12%, walk-in 3%, trap 2%, fixed-wing airplane 1%, genetic 0.3%, helicopter 0.1%; Table 4). For females, 98% of locations of founders and 99% Stirling-born fishers were within the Stirling boundary. The statistics for male fishers were 96% and 99%; 4 founding males took long forays after release, 3 without returning to Stirling [19]; the 1 who returned had gone further than 45 km from the edge of Stirling. Incidental detections of fishers on adjacent lands beginning in 2013 (Lassen National Forest, Lassen Volcanic National Park) indicated some male fishers survived beyond Stirling (California Department of Fish & Wildlife, unpublished data).

**Table 4. Mean numbers (+SD, N) of estimated locations per individual fisher per year across all years of study and 2017 organized by location method. Means are for individual fishers who were followed using each particular method. The research was conducted on or near the Stirling Management area owned by Sierra Pacific Industries and located in the Northern Sierra and Southern Cascade Mountains of Northern California.**

| Sex | All Locations | Triangulations | Walk ins | GPS | All Argos | Argos LC 2 + 3 |
|---|---|---|---|---|---|---|
| Female | 69 ± 82, 194 | 57 ± 48, 187 | 7 ± 6, 138 | 387 ± 259, 3 | | |
| Male | 304 ± 661, 88 | 16 ± 21, 21 | 2 ± 1, 8 | 1232 ± 1430, 11 | 175 ± 191, 73 | 36 ± 46, 73 |

*Population monitoring (Hypotheses 1, 2 and 3) --* After trapping in 2017, the fisher population size on Stirling estimated using spatial capture-recapture was 119 ± 96–141 [37] while the minimum number alive was known to be at least 68 (Fig 5). The estimates in 2017 were 81 females (48 of whom were adults) *vs* a minimum of 43 females known alive and 38 males (22 of whom were adults) *vs* 25 known alive. Both spatial capture-recapture and minimum numbers known indicated a growing population from inception through 2017. The age structure of the known fishers emphasized young fishers in all years except 2010 (Fig 7). On average the proportion of fishers <2 years old was 0.50 (range = 0.1–0.65; Fig 7).

As of 2017, most locations of fishers that we followed occurred within the boundaries of Stirling or very near to it (Fig 4). Similarly, most den locations have occurred on Stirling. At least 50 fishers remained on Stirling annually from 2014 through 2017, representing a core population that showed site fidelity (Fig 5).

*Home Ranges (Hypothesis 1) --* Table 5 shows mean size estimates for 95% utilization distributions for 2010–2017 using h = 750, 1000 and 1500 m. Males averaged larger utilization distributions than did females for all values of "h" and, logically, larger values for "h" yielded larger utilization distributions. Founding females established home ranges primarily within Stirling, as shown by [19], though some females travelled to adjacent Forest Service or private lands. One female traveled north 22 km onto the Lassen Management Area of Sierra Pacific; she died, however, within 3 months of release. Also, 15 female fishers denned in trees on both the Lassen and the Plumas National Forests (Fig 8). Male fishers established home ranges across much of Stirling. Since males had larger utilization distributions than did females and dispersed more widely, they were located on adjacent lands more often than were females. Some founding male fishers established home ranges off Stirling and up to 40 km from where they were released. We stopped tracking these fishers because their utilization distributions were outside the area that we trapped each year.

*Mortality (Hypotheses 2 and 3) --* We confirmed the deaths of 38 reintroduced and Stirling born fishers (27 F, 10 M, 1 unknown). One fisher kit found dead with its dead (presumed) mother in an abandoned water tank was too badly decomposed to determine sex. We

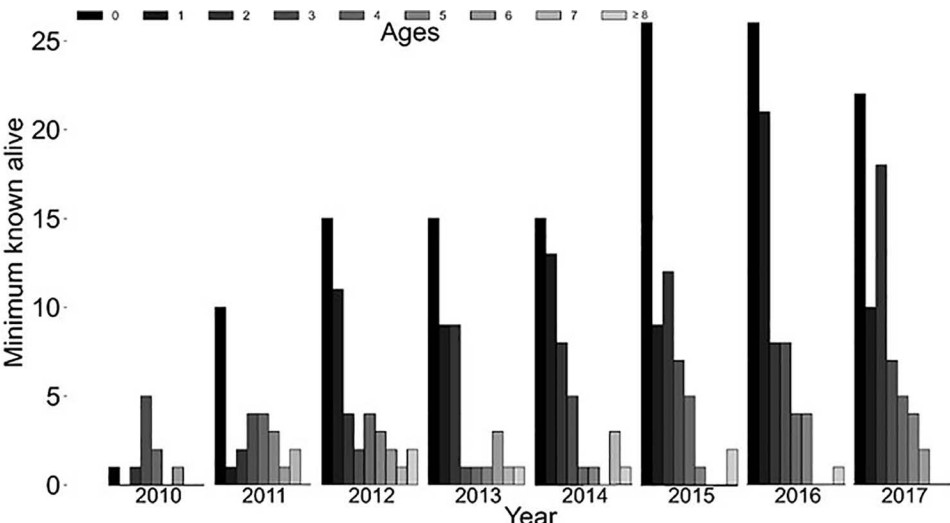

**Fig 7. Age distribution for all fishers known alive in each year from 2010 to 2017 on the Stirling Management Area of Sierra Pacific Industries in northern Sierra Nevada and southern Cascade Mountains of California.** Ages based on cementum annuli 2010-2017.

**Table 5. Mean areas (±SD) for 95% fixed kernel utilization distributions of fishers followed with telemetry for ≥ 6 months on the Stirling Management Area of Sierra Pacific Industries in the northern Sierra Nevada and the southern Cascade mounts of California 20109–2017. All utilization distributions were built using Silberman's k2. All female fishers followed were adults but 3 male fishers in 2012 and 1 in 2014 were juveniles.**

| Smoothing Parameter "h" (m) | Year | Mean Area of Utilization Distribution (km²), N | |
|---|---|---|---|
| 750 | 2010 | 17 ± 7, 6 | 67 ± 6, 3 |
| | 2011 | 28 ± 6, 7 | 114 ± 20, 3 |
| | 2012 | 17 ± 5, 12 | 56 ± 45, 9 |
| | 2013 | 15 ± 5, 13 | 46 ± 27, 3 |
| | 2014 | 16 ± 4, 13 | 63 ± 33, 3 |
| | 2015 | 15 ± 3, 19 | 40 ± 9, 3 |
| | 2016 | 12 ± 4, 19 | 25 ± 2, 3 |
| | 2017 | 13 ± 4, 19 | 67 ± 4, 4 |
| 1000 | 2010 | 22 ± 8, 6 | 97 ± 7, 3 |
| | 2011 | 37 ± 8, 7 | 143 ± 24, 3 |
| | 2012 | 22 ± 8, 12 | 75 ± 59, 9 |
| | 2013 | 18 ± 6, 13 | 63 ± 33, 3 |
| | 2014 | 19 ± 5, 13 | 77 ± 41, 3 |
| | 2015 | 18 ± 4, 19 | 57 ± 15, 3 |
| | 2016 | 14 ± 4, 19 | 33 ± 5, 3 |
| | 2017 | 15 ± 5, 19 | 90 ± 55, 4 |
| 1500 | 2010 | 32 ± 12, 6 | 153, ± 34, 3 |
| | 2011 | 56 ± 11, 7 | 189 ± 30, 3 |
| | 2012 | 30 ± 4, 12 | 108 ± 84, 9 |
| | 2013 | 24 ± 8, 13 | 94 ± 45, 3 |
| | 2014 | 25 ± 7, 13 | 100 ± 53, 3 |
| | 2015 | 23 ± 4, 19 | 88 ± 27, 3 |
| | 2016 | 19 ± 5, 19 | 40 ± 12, 3 |
| | 2017 | 20 ± 6, 19 | 132 ± 70, 4 |

documented the deaths of 18 (45% of 40) founding fishers, the last in August 2017 of a female born in 2007. When field research ended in 2017, 1 founding fisher was still alive, a male born in 2011. The average time from release until death was 21 ± 20 months. Causes of death included drowning in a water tank (n = 3), systemic disease of unknown origin (n = 2), killed by bobcat (*Lynx rufus*, 1), anticoagulant rodenticide (1), and road kill (1). Many carcasses showed potential evidence of predation but predation could not be documented conclusively. Eleven of 12 fisher carcasses tested had liver tissue test positive for exposure to anticoagulant rodenticides (92%), including brodifacoum, bromadialone, difethialone, chlorphacinone. Three of these 12 were founding fishers.

*Survival (Hypotheses 2 and 3)* – No models described survival better than the null hypothesis of constant survival, which ranked 4th among models tested (Table 2). The monthly survival estimate was 0.98 (0.97–0.98, 95% CI), making a yearly estimate of survival of 0.81 (0.74–0.86). Age described our data best, suggesting that survival decreased as age increased (Table 2). Reintroduction status (founder *vs* born on site) ranked second (ΔAICc = 1.73), suggesting that the founding fishers had lower survival than fishers born on Stirling. The model incorporating both age and the mean habitat value for fishers' locations ranked third, suggesting that survival increased with mean habitat quality. For all these models, slope could not be distinguished from 0 (= no effect). Thus, we can not reject the null hypothesis of constant survival.

Models that incorporated effects of habitat quality or the proportion of utilization distributions that were recently logged described our survival data very poorly (Table 2).

Our first estimate of 6-month survival for kits (based on the assumption that when females with kits died, their kits died also) was 0.80. Our second estimate (dividing the number of young-of-the-year captured in October and November divided by the number of kits documented at dens) was 0.94. Combining these estimates with 0.98 for monthly survival for kits' 6–12 months of age yields an annual survival rate of 0.67 for kits.

*Reproduction (Hypotheses 2 and 3)*– Fishers produced kits on Stirling every spring [46]. We located 262 dens (83 natal dens and 179 maternal dens), all of which except 1 were on or within 2 km of Stirling's boundaries (Fig 8). The mean denning rate for the entire study was 81% adult females denning per year, with 75% of denning females producing at least 1 surviving kit (Table 6) [46]. Most females moved their kits from natal dens to maternal dens, and to subsequent maternal dens, throughout spring and summer (Table 6). Female fishers chose cavities in black oaks most commonly for natal and maternal dens (50% of dens; Table 7). Female fishers chose cavities in live trees (*vs* snags) for natal dens most often (78%) but, later in the denning season as kits began to travel with their mothers, females more often chose dens in snags, hollow logs and piles of debris (47%). The denning season concluded by the end

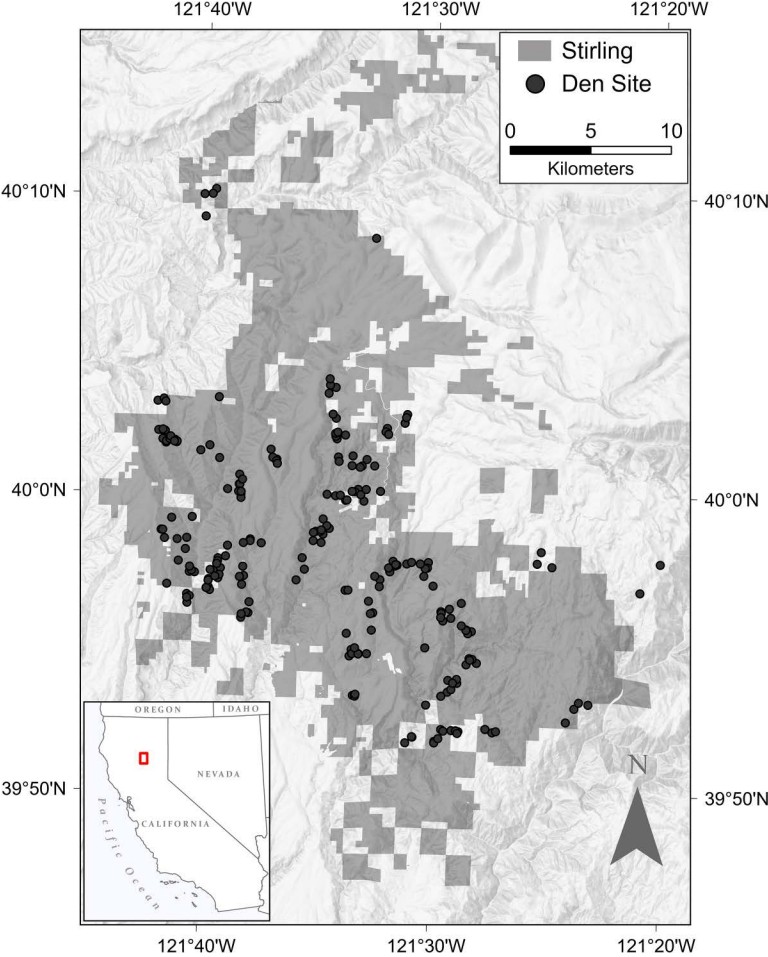

**Fig 8. Locations of fisher's dens on the Stirling Management Area of Sierra Pacific Industries in northern Sierra Nevada and southern Cascade Mountains of California, 2010-2017.**

**Table 6. Reproductive statistics for adult females that were radio-collared in 2010–2017 on the Stirling Management Area of Sierra Pacific Industries in the northern Sierra Nevada and southern Cascade Mountains of northern California.**

| Metric | 2010 | 2011 | 2012 | 2013 | 2014 | 2015 | 2016 | 2017 | Total |
|---|---|---|---|---|---|---|---|---|---|
| Females Tracked | 8 | 9 | 10 | 11 | 7 | 18 | 19 | 20 | 102 |
| Females Denned | 5 | 7 | 9 | 9 | 6 | 13 | 17 | 17 | 83 |
| % Denned | 63 | 78 | 90 | 82 | 86 | 72 | 89 | 85 | 81 |
| Min # Kits | 4 | 12 | 13 | 17 | 8 | 21 | 30 | 32 | 137 |
| Mean Litter Size | 1.0 ± 0 | 2.0 ± 0.5 | 1.9 ± 0.5 | 1.9 ± 0.2 | 1.6 ± 0.8 | 1.9 ± 0.4 | 2.0 ± 0.3 | 2.0 ± 0.2 | 1.9 ± 0.1 |
| Kits in Autumn | 1 | 10 | 15 | 15 | 17 | 26 | 27 | 22 | 133 |
| Kits Died in Dens | 2 | 3 | 3 | 2 | 2 | 1 | 6 | 2 | 21 |
| Kits/Female | 0.5 | 1.3 | 1.3 | 1.5 | 1.1 | 1.2 | 1.6 | 1.6 | 1.3 |
| Natal Dens | 5 | 7 | 8 | 9 | 4 | 13 | 16 | 17 | 82 |
| Maternal Dens | 12 | 11 | 16 | 16 | 1 | 30 | 42 | 56 | 183 |

**Table 7. Numbers of den trees by species for natal and maternal dens from 2010 to 2017, and by condition of the den tree (live tree, standing snag, or other [e.g., downed log or debris pile]) on the Stirling Management Area of Sierra Pacific Industries in the Northern Sierra Nevada and Southern Cascade Mountains of northern California.**

| | Natal | | | Maternal | | | |
|---|---|---|---|---|---|---|---|
| Tree Species | Live Tree | Snag | Other | Live Tree | Snag | Other | Total |
| Black oak (*Quercus kelloggii*) | 41 | 3 | 0 | 67 | 21 | 1 | 133 |
| Incense cedar (*Calocedrus decurrens*) | 6 | 7 | 0 | 5 | 25 | 0 | 43 |
| Douglas Fir (*Pseudotsuga menziesii*) | 4 | 1 | 0 | 7 | 16 | 0 | 28 |
| Unidentified conifer | 0 | 4 | 0 | 0 | 12 | 1 | 17 |
| Tanoak (*Notholithocarpus densiflorus*) | 6 | 1 | 0 | 5 | 0 | 2 | 14 |
| Canyon live oak (*Quercus chrysolepis*) | 3 | 0 | 0 | 7 | 0 | 0 | 10 |
| White Fir (*Abies concolor*) | 2 | 0 | 0 | 3 | 4 | 0 | 9 |
| Sugar Pine (*Pinus lambertiana*) | 1 | 1 | 0 | 2 | 2 | 0 | 6 |
| Ponderosa Pine (*Pinus ponderosa*) | 1 | 1 | 0 | 1 | 1 | 1 | 5 |
| Big Leaf Maple (*Acer macrophyllum*) | 0 | 0 | 0 | 0 | 1 | 0 | 1 |
| Total | 64 | 18 | 0 | 97 | 82 | 5 | 266 |

of August, after which most females moved kits often to rest sites, such as horizontal branches in trees, and juveniles started to become independent [26].

No model described reproductive rate better than the null hypothesis of constant reproduction (Table 8). The model that described our data best included only the proportion of a utilization distrib0ution logged within the previous 10 years; the reproductive output of female fishers decreased as the proportion logged increased. Nonetheless, the 95% credible intervals for the slope of this and all other models did not differ from 0.

Combining the results for survival and reproduction yields an estimate of population growth of λ =1.02. Our elasticity analyses show that variation in adult survival rate had greatest effects on population growth, followed by variation in 2nd year survival (Fig 9). Variation in adult survival also had the greatest effects on the extinction index (Fig 9).

*Habitat availability and selection (Hypothesis 4)*– Fishers selected large tree (closed canopy) forest stands in greater proportion than their availability and avoided stands with low canopy cover and small mean diameter trees (Fig 10).

Across all years, the mean available habitat quality across Stirling, calculated using Thomasma's model [36], was 0.44 (95% CI 0.42–0.47). The mean habitat value used by fishers was

**Table 8. Comparison of 10 Models with variables hypothesized to affect reproductive output by female fishers on the Stirling Management Area of Sierra Pacific Industries in the northern Sierra Nevada and southern Cascade Mountains of California, 2010-2017. Weights, *w*, for models Year and below have positive weights < 0.05 and round to 0.0.**

| Model | $AIC_C$ | $AIC_C$ | Likelihood | *w* |
|---|---|---|---|---|
| Proportion Utilization Distribution Logged | 285.1 | 0 | 1.00 | 0.8 |
| Year | 290.8 | 5.7 | 0.06 | 0.0 |
| Age | 292.6 | 7.6 | 0.02 | 0.0 |
| Habitat Quality | 292.7 | 7.7 | 0.02 | 0.0 |
| Proportion Hardwoods | 292.8 | 7.7 | 0.02 | 0.0 |
| Founding Fisher | 293.0 | 8.0 | 0.02 | 0.0 |
| Age + Year + AgexYear | 293.7 | 8.6 | 0.01 | 0.0 |
| Age + Habitat Quality | 294.4 | 9.3 | 0.01 | 0.0 |
| Founding Fisher + Age | 294.5 | 9.5 | 0.01 | 0.0 |
| Habitat Quality + Founding Fisher | 294.8 | 9.8 | 0.01 | 0.0 |

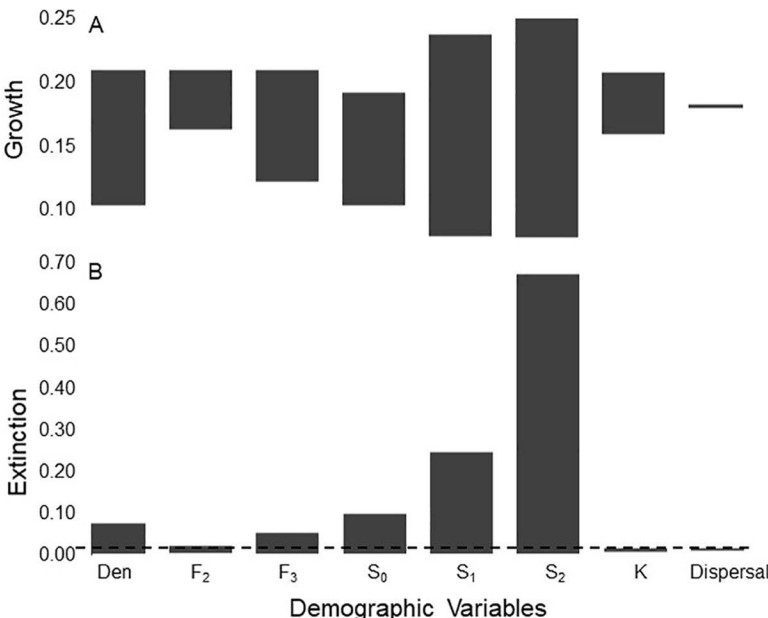

**Fig 9. Elasticity values for variables for a simulated population of female fishers.** Elasticity shows the effects on population growth rate and the effects on extinction of demographic variables representing life history traits. Baseline values for variables are the estimates for the Stirling fisher population. The horizontal dashed line shows the shows (A) the rate of population growth during the first 8 years of the reintroduction and (B) the extinction index, calculated as the proportion of 1000 simulations using the baseline values for demographic variables. The bars show the increases and decreases in population growth rate and the extinction index caused by 1-at-a-time 10% increases and decreases of the values of the demographic variables from baseline. Den = mean denning rate. F2 = mean number female offspring born to 2-year old females. F3 = mean number female offspring born to females > 3 years old. S0 = mean annual survival of female kits. S1 = mean annual survival of 1-year old females. S2 = mean annual survival of females > 2- years old. **K** = carrying capacity. Dispersal = dispersal rate.

0.62 (95% CI 0.60–0.65). The resource selection function regressed positively on the calculated habitat quality values (β = 0.42 [0.38–0.46], p < 0.001, $r^2$ = 0.84). Thus, collectively, fishers moved and rested sites that had high habitat quality and avoided areas that had low habitat quality (Figs 4, 10).

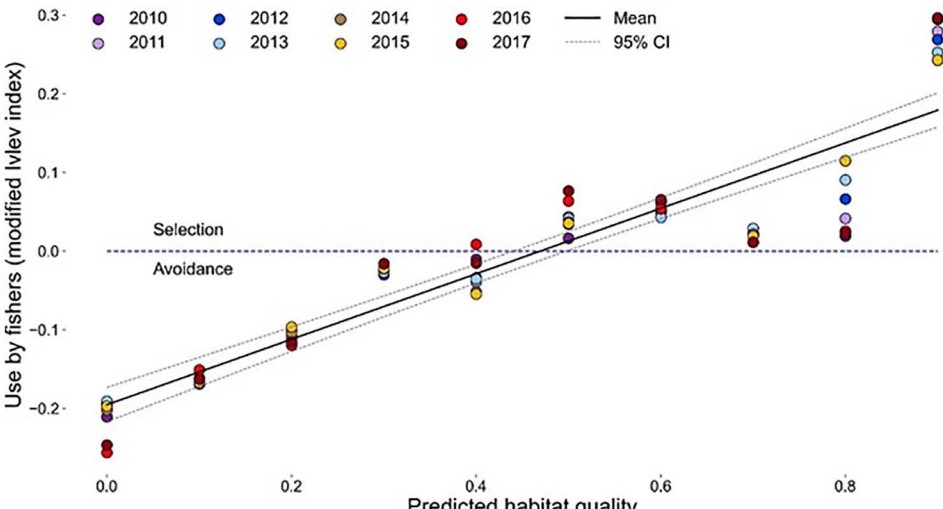

**Fig 10. Actual vs predicted use of land of different habitat quality, indexed by Thomasma's model, on the Stirling Management Area of Sierra Pacific Industries in northern Sierra Nevada and southern Cascade Mountains of California, 2010-2017.** The solid black line represents the mean ± 95% confidence intervals linear regression fitted for data in all years.

Habitat quality averaged across all female fishers' utilization distributions was 0.69 (95% CI 0.66–0.71). Male fishers had slightly lower mean habitat quality (0.56, 95% CI 0.51–0.61) and the mean of the means for each individual male's utilization distribution was 0.49 (95% CI 0.45–0.53). Habitat selection did not differ between the sexes (df = 1, $\chi^2$ = 0.29, p = 0.59), however, or among years (df = 7, $\chi^2$ = 5.59, p = 0.54). In all years, females selected habitat that was of significantly higher value than that available (Fig 11). The same was true for males in 6 of the 8 years of this research.

The proportion of utilization distributions that included recent clearcuts did not differ between females (11%, 95% CI 10–13) and males (14%, 95% CI 11–17; df = 1, $\chi^2$ = 0.01, p = 0.944) but did correlate positively with year ($\beta$ = 0.106 [0.004–0.208], $\chi^2$ = 4.17, p = 0.041; Fig 12). We found no interactions between sex and year (df = 1, $\chi^2$ = 0.01, p = 0.946).

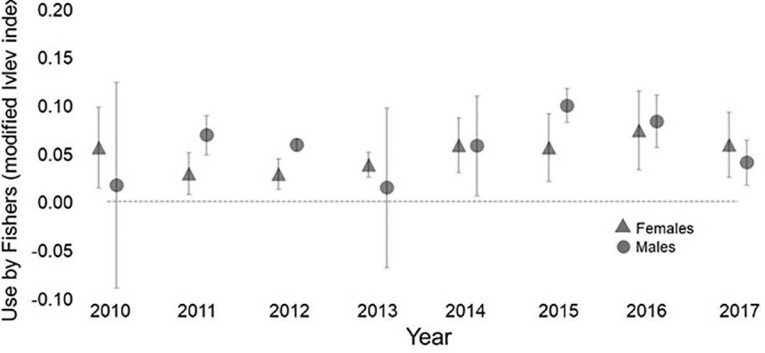

**Fig 11. The mean modified Ivlev resource selection values ± 95% CI for male and female fishers by year for all fishers with at least 20 locations in a year.** For each fisher, the resource selection value is the mean value from all locations in a year. Data are for fishers living on the Stirling Management Area of Sierra Pacific Industries in northern Sierra Nevada and southern Cascade Mountains of California, 2010-2017.

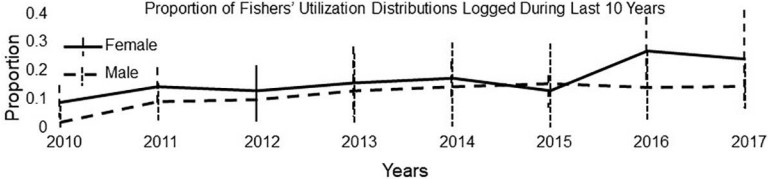

**Fig 12. The mean proportion ( ±SD) of fishers' utilization distributions subject to timber harvest in the preceding 10 years.** Proportions for female and male fishers do not differ but the combined proportions increase significantly over time. Data for fishers on the Stirling Management Area of Sierra Pacific Industries in northern Sierra Nevada and southern Cascade Mountains of California, 2010-2017.

*Population Simulations and Viability Analyses*– Our Baseline simulations produced unsurprising outcomes (Fig 13). The mean simulated population size in all simulation runs grew for about 10 years, reaching a little over 40 females, after which time population size varied just under *K*. The extinction index for the population was 0.01. A wildfire in year 10 caused fisher population decrease and the decrease was greater with greater percentage of the area burned (Fig 13). The simulated population continued to decrease for about 10 years after the fire while the simulated landscape had a large area of early seral forest. After 10 years, much of the burned area was modelled to have succeeded to forest with small trees with a nearly closed canopy and the simulated fisher population increased gradually until the forest succeeded to larger, more open forest, which caused the population to dip again. The extinction index for the simulated population in the Baseline scenario increased progressively more steeply with increasingly larger burns, and exceeded 0.8 when fire size was 60% of the simulated landscape (Fig 14).

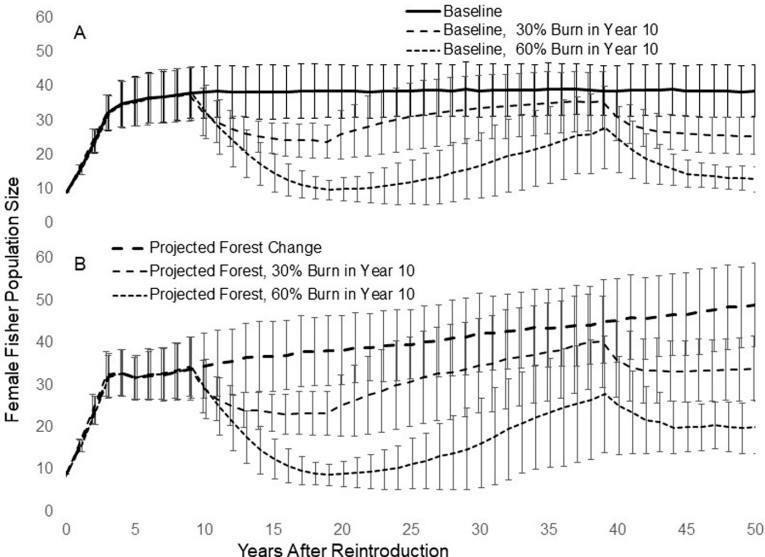

**Fig 13. Stochastic population simulations for female fishers living under 2 sets of scenarios.** The simulations are based on the fisher population reintroduced to Stirling. A. Baseline. Simulation with no changes in habitat quality, reproductive rate, survival rate or carrying capacity (K). Also, simulations of population change after a wildfire burns 30% or 60% of the simulated landscape in year 10. B. Simulation with habitat quality changing as proposed by Sierra Pacific, leading to changes in *K* concomitant with changes in vital rates and dispersal. Also, simulations of population change after a wildfire burns 30% or 60% of the simulated landscape in year 10.

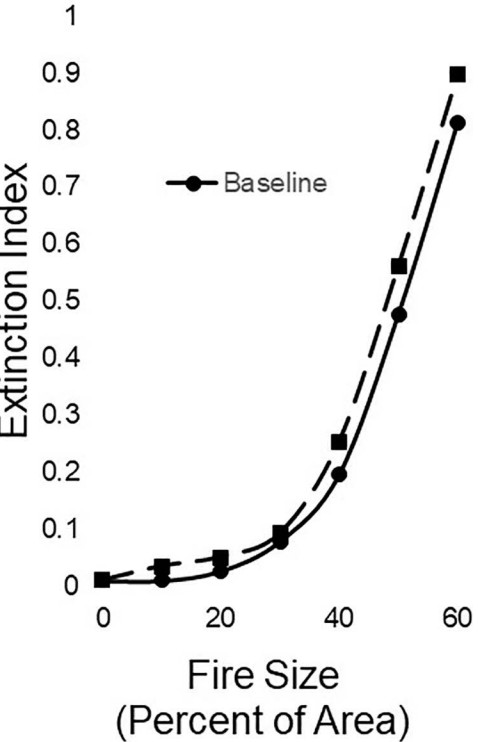

**Fig 14. For a simulated population of female fishers, changes in the extinction index (proportion of simulation runs that led to extinction) with the extent of a wildfire on a simulated landscape.**

Our Baseline simulations suggested that juvenile survival rate has large effects on fisher population growth (Fig 15). All simulated populations grew rapidly during the first 3 years, while releases were ongoing, and no simulated population went extinct before year 7, even at the lowest rate of juvenile survival. Simulated reintroductions that strongly decreased or

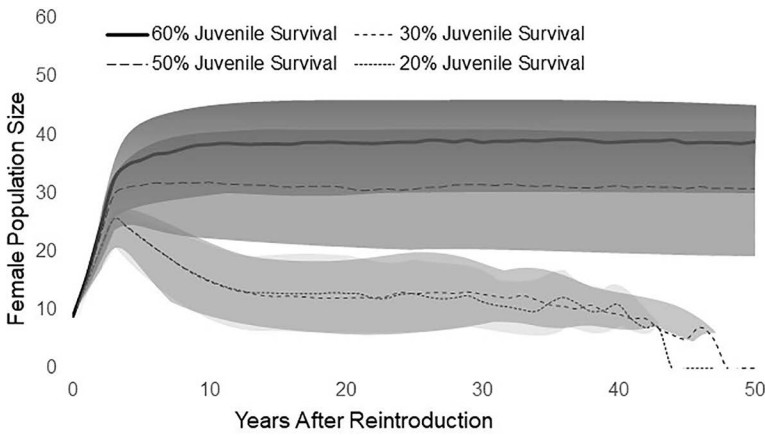

**Fig 15. Projections of mean female population size (from 100 simulated translocations) with the mean values ± 1 standard deviation (shaded portions) for simulated populations where mean juvenile (kit) survival was 20% (dotted line), 30% (small dashes), 50% (large dashes) and 60% (solid line).**

increased could not be differentiated clearly before year 5 because of the influence of the releases during the first 3 years and large variations in population growth.

When we set mean juvenile survival to 0.6, which was less than the estimate suggested by our data, the mean female population size grew to carrying capacity ($K = 45$) within about 10 years and only 10 of 1000 simulated populations went extinct, yielding an extinction index of 0.01 (Fig 15). At the other extreme, when we set mean juvenile survival to 0.2, a value taken from the literature [92,93], all populations went extinct by 44 years, with the earliest extinction at 8 years. For a mean juvenile survival rate of 0.30, all populations went extinct by 48 years. For juvenile survival of 0.50, the population mean grew to about 40 female fishers and stabilized, with an extinction index of 0.25 (Fig 15).

In our timber Harvest simulations, the simulated fisher population grew rapidly following reintroduction but then dipped when the population reached about 35 female fishers at 5–6 years. At this time, the simulated population no longer received supplementation from new releases while the proportion of the simulated forest in early seral stage grew (Fig 13). Thereafter, the fisher population grew gradually to shy of 50 females as more of the forest reached the stages with large trees. As with the Baseline scenario, the extinction index for the population was about 0.01.

A simulated forest fire in year 10 caused the population to decrease for about 10 years post-fire, after which much of the burned area had succeeded to forest with small trees with a nearly closed canopy (Fig 13). Thereafter the simulated fisher population increased gradually but to higher population sizes than seen in the Baseline simulations. When the forest developed to larger, more open stages of managed forest, the simulated fisher population dipped again. The extinction index was 0.01 with no wildfire and increased progressively more steeply nearly to 0.9 when fire size was 60% of the simulated landscape (Fig 14).

## Discussion

Our project provides a case study of a reintroduction where organisms were released according to a detailed protocol that incorporated good research design. We assessed habitat before and during monitoring and evaluated habitat use; we detailed mortality, survival and reproduction with respect to population growth and stability; and we documented home ranges and site fidelity. In the end, we established a new, functioning population. We collected pertinent data to document why the new population became established but also so that we could have explained why a new population did not become established, had that been the outcome. In addition, we tested several hypotheses related to the biology of the target species, aided by the control we had over the reintroduction: initial population size and sex ratio can be controlled [21; this study]; habitat and food can be quantified [18,60,61]; appropriate methods for monitoring the new population and source populations can be adopted [37,62]; and effects on resident populations of other animals and plants can be monitored [18,61].

*Home Range Establishment* -- The responses of fishers to being released onto Stirling, specifically their site fidelity after release (Hypothesis 1; Table 1), is an important measure of how they perceived their environment and its habitat quality upon release [58]. Both founding and Stirling-born animals settled developed utilization distributions within 500 days after release [19] and the population grew through 2017, when field research ended (Fig 4). We reject the null hypothesis for Hypothesis 1 (no site fidelity, Table 1) since fishers did exhibit site fidelity on land managed for timber production.

The sizes of utilization distributions that we obtained for h=750 are similar to those reported in the review by Lofroth et al. [50]. Unfortunately, since published research has

specified neither h nor the kernels used for building utilization distributions for fishers, we can not know whether sizes of utilization distributions reported emphasize mean daily movements, the radius of movements, maximum movements, or some other aspect of fisher biology.

*Survival and Reproduction* -- A main goal for our research was to understand how habitat and forest management practices affected fisher survival and reproduction on Stirling (Hypotheses 2, 3; Table 1). With an 8-year study, we have confidence in our estimates of vital rates and population growth [55] and the causes of mortality that we documented are consistent causes found elsewhere in California [40,41,66,102]. We documented that many fishers had been exposed to anticoagulant rodenticides. The impacts on fishers and other wildlife of sub-lethal exposure to anticoagulants is largely unknown, though widespread exposure and cases of direct mortality of fishers and other wildlife has raised conservation concerns [66,87]. Because some fishers born on Stirling tested positive for rodenticides, those fishers must have been exposed to rodenticides on Stirling. Sierra Pacific did not use rodenticides for any forest management practices, suggesting that exposure to rodenticides was from secreted marijuana grow sites on Stirling [66,87].

We found little evidence of strong differences between the sexes in survival, for effects of habitat, or for effects of being moved for reintroduction. Thus, we reject the null for Hypothesis 2 (Table 1) and accept that survival and reproduction of fishers on Stirling were adequate to establish a new population.

We can not reject the null hypothesis for Hypothesis 3, which was that logging did not affect survival and reproduction negatively. Recent logging and reforestation (leaving early seral forest) within female fishers' home range on Stirling appears not to have reduced reproductive output nor decreased survival through 2017, at least at the levels of timber harvest that fishers experienced. That our population grew through 2017 limits our ability to quantify at what level logging affects carrying capacity of fishers on Stirling negatively. Nonetheless, by 2017 our fisher population had surpassed our estimate of carrying capacity. Our inability to document effects of logging on reproduction may have been limited by the narrow range of habitat quality available. Had our field research continued, we may have documented effects of forest changes (positive or negative) on reproduction. Even so, we did document that fishers on Stirling reproduced at rates capable of maintaining a viable population. These results suggest that habitat and prey availability did not limit survival and reproduction in our new population during its Establishment Phase on an industrial, managed timberland.

Our estimate of kit survival for the first year, 0.67, is distinctly higher than the values of 0.2–0.25 reported elsewhere [93,103]. Other researchers estimated litter size at birth by counting corpora lutea, which can vastly overestimate numbers of kits born for mammals with long delayed implantation, as fishers have [26,42,104,105]. A female in a good year might give birth to and raise the maximum litter size (matching the number of corpora lutea), while a female in a very bad year might not bring any of the eggs shed through to birth (distinctly not matching the number of corpora lutea). We interpret these results of kit survival to mean that previous estimates were biased extremely low because litter size at birth was vastly over-estimated. We find our estimate of 0.67 a trustworthy estimate of kit survival for our study.

*Population Growth* -- By 2014, mating, reproduction, and recruitment were no longer restricted predominantly to the founding fishers and the population had increased to an estimated size that was larger than the number of founding fishers. The vital rates observed on Stirling in 2017 yield $\lambda$ =1.02. This value for $\lambda$ is consistent with our simulations for population growth for populations with high levels of kit survival (Figs 7, 13, 15). In addition, in 2017 the young age structure of the population suggested healthy reproduction and recruitment (Fig 7). We reject the null hypothesis for Hypothesis 3 and retain the hypothesis that fishers

on timberland managed as Sierra Pacific managed Stirling can maintain vital rates sufficient for population growth during the Establishment Phase. Therefore, the fisher population succeeded through its Establishment Phase and into its Persistence Phase (Fig 1).

Our estimate of $K$ for female fishers, 43 females in 2017, was distinctly lower than our estimate of the population size using spatial capture-recapture models but similar to the number of adult females (81 total females, 48 adults) [37,46] for a total of 199 fishers. The estimate for the numbers of males was 38 (spatial capture-recapture). Spatial capture-recapture models have the advantage of being able to calculate credible intervals for population sizes each year and this could have placed the female population as low as 62.

The 119 fishers of both sexes yields a density of about 10.8 fishers/100 km². This population density is very similar to the range of densities observed for other fisher populations in California and across western North America: 5–52 fishers/100 km² [26,30,37]. The density of 52 fishers/100 km² was in 1998 on the Hoopa Reservation where fishers occupied largely old growth forest, which was high quality habitat [30]. That population, however, had dropped to 14.7 fishers/100 km² by 2005. Densities of 25–40 fishers/100 km² in eastern North America were in areas of largely unbroken forest where fishers had never been extirpated [23,24,106–109]. The density of fishers on Stirling is also consistent with the densities for fishers across much of their present range, where habitats are generally considered to be of moderate quality but capable of maintaining viable populations (Table 3) [26,28]. Nonetheless, fishers living in these modest habitats exhibit significant preference for the best available habitats, as defined by Thomasma's model (Fig 10).

Our analyses focused on broad population processes that likely represent the overall habitat conditions in the study area [95,96]. The forest characteristics selected by fishers included relatively mature forests with canopy and shrub layer structuring and with moderate percentages of hardwoods (Fig 4). Our female fishers selected natal dens in black oaks, underlining the importance of the hardwood component of Thomasma's habitat model. For all these analyses, however, we weighted all location estimates equally, assuming that the time that fishers spend in a place indexes that place's importance. Additional habitat modeling may show other metrics that help to explain fishers' habitat selection.

*Habitat Selection* -- Fishers in the first few years of our research established home ranges with only small proportions that had been logged recently but those proportions increased through time (Hypothesis 4; Table 1). As the fisher population grew, the number of clearcut stands in the study area increased and some clearcuts were inside established utilization distributions of fishers. Fishers avoided clearcut stands (Figs 10, 11; they rank of low quality in Fig 4) and the mean proportion of fishers' utilization distributions in clearcuts remained below 15%, which may be a threshold for how much clearcut area a fisher will tolerate [26,28–31,33]. Such thresholds have been documented for American martens (*Martes americana*) [33,110–113].

Thus, we conclude that forest management for timber production leading up to and during our research on Stirling provided sufficient forest cover with mature forest characteristics and adequate habitat for prey species (*e.g.*, gray squirrels [*Sciurus griseus*], which require trees that produce large seeds) to allow a new fisher population to become established, albeit at a low density. We reject the null for Hypothesis 4 because fishers did select mature forest habitats. Fishers were able to select enough forest with mature characteristics to establish a population.

*Population Simulations* -- Our population simulations explored how the fisher population on Stirling was expected to respond to changes in forest conditions, given what we understand about fisher biology. All simulated populations grew rapidly while releases were ongoing and no simulated population went extinct prior to year 7, providing additional support for our decision to monitor the real population for 8 years. The extinction index was consistently

lower for the simulations with the forest managed for change without fire, consistent with the goals of Sierra Pacific for Stirling, and with its management strategy to use even aged management. We added simulated fires, however, in year 10, when the forest had its largest proportion in the early seral stage. Had we simulated fires to occur later when the extent or early seral stage forest was smaller, our simulations may have produced somewhat smaller extinction indices (Fig 14).

Elasticity measures the effects on model output from small changes in values of variables and large elasticity values for a particular variable suggest that values of that variable have large effects on model output. Adult survival, 1 year- old survival, and denning rate had the largest elasticity values related to population growth, and adult survival and 1 year- old survival had the largest elasticity values for the extinction index. Elasticity results, however, are also affected by model structure. For example, consistent with Bulmer's [100,101] analyses, we reduced age-specific survival rates as population size increased but not age-specific litter sizes. Thus, the model structure dictated that changes in survival would have greater effects on survival than would changes in reproduction.

In addition, we built a stage-specific Leslie matrix and not an age-specific matrix. The consequence is that adult survival was effectively repeated at each time step by as many adult age classes as existed within the population. Had we built an age-specific model (as suggested by the potential effects of age on survival) extending through the 10-year age span for fishers, each older age-specific survival rate would have had progressively smaller and smaller effects on population growth, leaving $S0$ (survival to age 1) and $S1$ (survival from age 1–2) as the age-specific survival rates with the largest effects on population growth. Nonetheless, juvenile survival in our model had a smaller influence on population growth than did survival of yearlings and adults, which contrasts with the models developed by Buskirk et al. [93] and Lewis et al. [52]. Their models used the lowest juvenile survival rates that we used, which increased the relative importance of juvenile survival. As we highlighted above, strong evidence suggests that those low rates of juvenile survival were calculated in error. We studied a population that was becoming established, with animals presumably experiencing little competition, especially during the early years of our study. Consequently, the high survival rates that we calculated for first-year fishers may have depended on low population densities. These results highlight the potential importance of the juvenile cohort and the very real importance of the lack of good estimates for juvenile survival.

Finally, our model included litter size as a stage-specific rate, as with adult survival. Consequently, the effect of F3 (adult fecundity) on population growth is inflated compared to age specific litter sizes in an age-specific matrix model. Likewise, had our model not allowed different litter sizes for first and subsequent litters, then the importance of litter size would have been inflated. In the end, our elasticity values must be considered within the context of our model.

Using *Vortex* software (SCTI, Brookfield, Illinois) to build an age-specific, spatially specific model for a fisher reintroduction, Lewis et al [31] found that litter size and juvenile survival had the greatest effects on reintroduction failure through population extinction. *Vortex* includes some realistic population effects, such as an Allee effect at small population sizes, that we did not incorporate into our model. Consequently, our model may underestimate extinction probabilities. In contrast, however, our model allowed juvenile dispersal beyond the simulated study site, and recolonization of our simulated study site from the surroundings, making our simulated population an open population. Allowing dispersal and recolonization provided our model population with greater resiliency to extinction.

*Final Thoughts* -- The fisher population on Stirling was isolated and relatively small but had moved through the Establishment Phase and into its Persistence Phase. Only future research can determine whether the local population continues to be viable and whether carrying capacity will change in the future. By 2017, the population was expanding onto

neighboring USDA Forest Service and private lands. Forest management on Stirling influenced fishers by altering the density of cavities for reproduction, the habitat conditions for prey, and habitat conditions for competing predators, all known to be important for fishers [26,29,30,38,39]. Habitat changes through forest management affect placements of fishers' home ranges, fitness of individual fishers and, ultimately, population sizes [38]. We hypothesize that fisher densities on Stirling and on the neighboring lands will correlate with how well forest management practices on each property maintain high canopy cover, denning trees (especially old hardwoods and snags), and habitat for prey species, especially western grey squirrels, and maybe even porcupines.

Re-establishing a population should not be the sole goal of a reintroduction [16,18,20,114–116]. We have demonstrated that experimental reintroductions can be used to test the prevailing knowledge of an organism's habitat, which is particularly important when disagreement exists regarding what constitutes high-quality habitat [14,18,56]. Had most founding fishers dispersed from Stirling, we might infer that habitat or prey populations were insufficient to support a fisher population [57,58]. Alternately, had most fishers not dispersed but died quickly or failed to reproduce, we might also infer that habitat quality or prey populations were insufficient both on and off Stirling (Fig 1) [117,118].

Fishers are particularly interesting because they are specialists that exhibit flexibility. Fishers are clearly specialized to prey on porcupines [26,28,119] a prey for which they have no effective competitors. Yet, they can maintain populations in areas effectively lacking porcupines, such as Stirling. Similarly, fishers appear specialized for continuous, mature forests yet can maintain populations in second growth forests. Indeed, their flexibility related to forests has increased in recent decades, as fisher populations have moved into wooded, suburban areas [32,120]. In the mid-1900s, fishers consistently used continuous forests and avidly avoided any open areas, avoided humans, and were highly secretive [23,24,106,121]. An hypothesis explaining this behavior is that the incredibly strong selective pressure from trapping and hunting fishers for fur in the late 1800s and early 1900s that reduced fishers to 6 small populations in the US [26] selected behaviors that minimized fishers' exposure to humans. Before European colonization, fishers appear to have specialized on mature and old growth forests [122–124]. Such forests were patchy and included open areas and early successional forest patches caused by drained beaver meadows, windthrow and fire. Fishers might have hunted in those habitats to some extent. With the relaxation of intense selection since the 1940s, fishers may have responded to selection to use more diverse habitats once again, such as early regenerating forest stands.

Fishers provide the opportunity to test hypotheses related to the influences of climate and forest management practices on wildlife. Fishers have fared poorly in western North America compared to fishers in eastern North America [31,52,62,120] where fishers experience high rainfall, low temperatures, moderate snowfall of high density snow, high primary productivity, and relatively few predators. Those conditions also exist where fisher have high densities in western North America [29,30,125,126]. Nonetheless, in many places in western North America, good fisher habitat is limited to ribbons of forest at intermediate elevations wrapping mountain ranges. Such elevational ribbons limit population expansion and climate change may cause ribbons to shrink or disappear as they move upslope. Few studies have looked at the effects of climate and other abiotic factors on the fisher's range. We hypothesize that climate and other abiotic conditions affect regional and local fisher populations through direct effects on fisher behavior.

Our research provides a narrow test of the hypothesis that a forested landscape managed in accordance with the California Forest Practice Rules can provide suitable habitat for fishers. Sierra Pacific also adopted a Candidate Conservation Agreement with Assurances with the

U.S. Fish and Wildlife Service and had its own voluntary management policies, which provided habitat for fishers beyond that required by the Forest Practice Rules. We can not know how much difference those extra policies made, so our test of the hypothesis is narrow. On Stirling, the fisher population persisted within a management area that included numerous and ongoing small clearcuts (± 8 ha each) that constituted in total about 25% of the area (Fig 3). This persistence is consistent with habitat flexibility shown by other fisher populations. Fishers appear able to maintain populations on fragmented or patchy landscapes [26,28,30] and even in urban environments. Our study documents that fishers are adaptable and capable of finding pockets of suitable habitat even if they are not ideal.

We do not know in what ways the Stirling fisher population will change, or how fishers will respond to forest management on Stirling. Our data support the hypothesis that fishers can live on heterogeneous landscapes like Stirling managed for timber production, though the threshold for the degree of fragmentation due to logging remains unknown [28,38,39]

Our research provides insight into the merits of reintroducing organisms into areas with less than optimal habitat, which may incorporate good and not so good habitat on a landscape. A strategically sited, new population, even if its population density is low, can lead to colonization of further new areas, expanding the conservation value of the reintroduced population. In addition, having one more population of a species that is of conservation concern increases resilience of the species to extinction. A reintroduction that includes well designed research can discover important new information on the biology of a species.

## Acknowledgements

Many groups provided funding, logistic support, and technical assistance. The California Department of Fish and Wildlife, U.S. Fish and Wildlife Service, Sierra Pacific Industries, and North Carolina State University are the 4 key cooperators and were responsible for carrying out the research. Assistance in locating and monitoring den sites was provided by Talbert Alvarado, Jason Banaszak, Colin Beach, Jessica Bodle, Phillip "Mike" Caulder, May Dixon, Amy Fontaine, Jesse Hogg, Pierce Holland, Dustin Marsh, Laura McMahon, John Morris, Matt Reno, Khris Rulon, Michelle Schroeder, Julie Shaw, Rob Swiers, Mary Talley, Andria Townsend, and Isaiah Williams. Robert Carey (U.S. Fish and Wildlife Service), Scott Hill (California Department of Fish and Wildlife), Cajun James, Steve Roberts, and Dennis Thibeault (Sierra Pacific Industries) provided support and contributions throughout the project. Logistical and trapping support, access to land, and other important contributions were provided by USDA Forest Service (Lassen and Plumas forests).

## Author contributions

**Conceptualization:** Roger A. Powell, Richard Callas, Ed Murphy.

**Data curation:** Roger A. Powell, Aaron N. Facka, Kevin P. Smith, Sean M. Matthews.

**Formal analysis:** Roger A. Powell, Aaron N. Facka.

**Funding acquisition:** Roger A. Powell, J. Scott Yaeger, Pete Figura, Richard Callas, Ed Murphy.

**Investigation:** Roger A. Powell, Aaron N. Facka, Deana L. Clifford, Kevin P. Smith, Sean M. Matthews, Pete Figura, Ed Murphy.

**Methodology:** Roger A. Powell, Deana L. Clifford, Richard Callas.

**Project administration:** Roger A. Powell, Sean M. Matthews, J. Scott Yaeger, Pete Figura.

**Resources:** Roger A. Powell, Deana L. Clifford, J. Scott Yaeger, Pete Figura, Richard Callas, Ed Murphy.

**Software:** Roger A. Powell, Aaron N. Facka.

**Supervision:** Roger A. Powell, Aaron N. Facka, Deana L. Clifford, Kevin P. Smith, Sean M. Matthews, Richard Callas.

**Validation:** Roger A. Powell, Aaron N. Facka.

**Visualization:** Roger A. Powell.

**Writing – original draft:** Roger A. Powell, Aaron N. Facka.

**Writing – review & editing:** Roger A. Powell, Aaron N. Facka, Deana L. Clifford, Kevin P. Smith, Sean M. Matthews, J. Scott Yaeger, Pete Figura, Richard Callas, Ed Murphy.

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
