## [Decision Letter · Decision Letter 0]

17 May 2024

PONE-D-24-04811Establishing a Late Successional Carnivoran on an Intensively Managed Landscape Reintroducing Fishers and Doing SciencePLOS ONE

Dear Dr. Powell,

Thank you for submitting your manuscript to PLOS ONE. After careful consideration, we feel that it has merit but does not fully meet PLOS ONE’s publication criteria as it currently stands. Therefore, we invite you to submit a revised version of the manuscript that addresses the points raised during the review process.

This paper evaluates the potential influence of habitat quality on the reintroduction of fishers, a carnivore of conservation concern. It will make an important contribution to our understanding of how to improve reintroduction efforts in managed lands, but the reviewers and I believe it needs a little more clarification, some streamlining, potential reorganization of the sections of the paper, and some additional context. Both reviewers have identified some specific areas that were unclear or need some revision.  Please note they have provided thorough line-by-line suggestions that will help improve the flow and strength of the paper.  I would also recommend trying to streamline the paper as much as possible and consider some Figures/Tables (e.g., Figures 8 & 9), while interesting, might be suitable for a supplement to highlight the most critical findings in the main text. The abstract is a bit vague on the habitat quality aspect of the study and Line 42 is confusing as written.  I agree with the reviewers that the hypotheses presented in the introduction are a good framework that could be used throughout the paper to organize the methods, results and discussion in the same sequence.  As noted by Reviewer 1, reorder the hypotheses if you would prefer a different order but it would help the reader to have them always presented in a consistent order.  Reviewer 2 notes that some of the hypotheses may not be explicitly tested in this paper and there is a potential 7^th^ hypothesis in the methods, while Reviewer 1 suggests a 7^th^ hypothesis related to fire.  There is good detail in the methods, but some aspects are highlighted by the reviewers that need some additional explanation/clarification (e.g., time frame for cover classes).  Line 350 is missing parentheses in denominator of equation. The results are very thorough, but could also use some additional clarification in places as noted by the reviewers and both offer suggestions on improving the figures and tables.  Table 1 may not be needed, though, it could be modified to be more useful with an indication of whether or not each is supported by the data presented.  The discussion could be strengthened with some reorganization, especially in relation to the proposed hypotheses. Reviewer 2 also makes some interesting suggestions on aspects to consider for the discussion that will add additional context and strengthen the broader applicability by considering the larger landscape. This paper has a lot of potential but needs some revision before suitable for publication.

We look forward to receiving your revised manuscript.

Kind regards,

Karen Root, Ph.D.

Academic Editor

PLOS ONE

Journal Requirements:

 [The California Department of Fish and Wildlife, U.S. Fish and Wildlife Service, Sierra Pacific Industries, and North Carolina State University are the 4 key cooperators and were responsible for carrying out the research and funding the reintroduction.].  

[Many groups provided funding, logistic support, and technical assistance. The California Department of 

Fish and Wildlife, U.S. Fish and Wildlife Service, Sierra Pacific Industries, and North Carolina State 

University are the 4 key cooperators and were responsible for carrying out the research and funding the 

reintroduction. Assistance in locating and monitoring den sites was provided by Talbert Alvarado, Jason 

Banaszak, Colin Beach, Jessica Bodle, Phillip “Mike” Caulder, May Dixon, Amy Fontaine, Jesse Hogg, 

Pierce Holland, Dustin Marsh, Laura McMahon, John Morris, Matt Reno, Khris Rulon, Michelle 

Schroeder, Julie Shaw, Rob Swiers, Mary Talley, Andria Townsend, and Isaiah Williams. Robert Carey 

(U.S. Fish and Wildlife Service), Scott Hill (California Department of Fish and Wildlife), Cajun James, 

Steve Roberts, and Dennis Thibeault (Sierra Pacific Industries) provided support and contributions 

throughout the project. Logistical and trapping support, access to land, and other important 

contributions were provided by USDA Forest Service (Lassen and Plumas forests)]

  [The California Department of Fish and Wildlife, U.S. Fish and Wildlife Service, Sierra Pacific Industries, and North Carolina State University are the 4 key cooperators and were responsible for carrying out the research and funding the reintroduction.]. 

5. Please update your submission to use the PLOS LaTeX template. The template and more information on our requirements for LaTeX submissions can be found at http://journals.plos.org/plosone/s/latex.

6. Please amend the manuscript submission data (via Edit Submission) to include author Tom Engstrom and Laura Finley.

Reviewers' comments:

Reviewer's Responses to Questions

**Comments to the Author**

1. Is the manuscript technically sound, and do the data support the conclusions?

Reviewer #1: Yes

Reviewer #2: Yes

2. Has the statistical analysis been performed appropriately and rigorously? 

Reviewer #1: Yes

Reviewer #2: Yes

3. Have the authors made all data underlying the findings in their manuscript fully available?

Reviewer #1: No

Reviewer #2: Yes

4. Is the manuscript presented in an intelligible fashion and written in standard English?

Reviewer #1: Yes

Reviewer #2: Yes

5. Review Comments to the Author

Reviewer #1: Summary:

This manuscript is a sort of capstone monograph describing the lessons learned from the final decade of the lead authors’ career before transitioning to emeritus (where he has continued to be productive in retirement). In this manuscript, the authors document the reintroduction of fishers into land in California that is managed for timber. Monitoring the reintroduced population for 8 years showed promising monthly survival and reproduction rates. The population remained small (<70) but grew steadily throughout the study. The authors performed simulation modeling and predicted that the population is unlikely to go extinct over the next 40 years. This study is a very thorough consideration of the feasibility of reintroducing fishers into less-than-ideal habitat, as well as a broader exploration of the value of using reintroductions to assess resource use and population dynamics of a species. The recommendation that “non ideal” habitat be considered for fisher reintroductions is well supported. There is also strong support for the authors’ argument that re-establishing a population should not be the sole goal of a reintroduction. This manuscript covered complex concepts in mostly excellent, clear prose. This is a solid, thorough manuscript–my quibbles are mostly minor and are listed below.

Minor comments:

Line 42: I think the authors mean "wildfire" instead of "wildlife" here. If so, they should correct the text to prevent potential confusion.

Line 67: The argument that ease of access or detailed habitat information compensates for lower habitat quality needs to be qualified. This can contribute to the ability of the habitat to make a great study site, but it does not contribute to the ability of the habitat to support the species.

Line 83: “and are” rather than “are”.

Lines 91-93: Some kinds of grouse (e.g., ruffed) and snowshoe hares prefer, or at least do fine in, early-successional forest. I do not think it is accurate to say that these are mature-forest species, so I would recommend deleting this sentence.

Line 94: The in text citation “Buskirk and Powell 1994” seems to refer to the reference in line 1169 (a chapter authored by Powell from a volume written by Buskirk, Harestad, Raphael, and Powell). If so, the in-text citation should read “Powell 1994.”

Line 115: It is unclear what “close genetic relatedness” means in this context. Some clarification would be helpful.

Line 120-137: These paragraphs contain background that is arguably not necessary and could be cut or severely reduced. If the authors feel strongly that it is necessary information, it would probably be better suited for the Materials and Methods section, under the “study site” subheading.

Lines 158-172: The rationale for these hypotheses is not always clear. They would benefit from some extra detail in a “Because ABC, then we hypothesized XYZ” construction.

Line 165: It is unclear what “moved before 1 January” means. The authors could perhaps clarify by saying “translocated before 1 January of a respective year.”

Line 173: The authors should consider adding the effects of fire as a 7th hypothesis; the current paragraph is underdeveloped and feels like an afterthought.

Line 180: “Field Methods” implies another Methods section, perhaps titled “Analytical Methods”. No such section appears to exist. Therefore I recommend changing the title at line 180 to “Materials and Methods” or “Methods” and then using subheadings, if desired, for field and analytical methods.

Lines 181-209: Even though the authors use past tense throughout the paper, in this section they are describing an existing landscape. It would make more sense to the reader if they were to describe it in present tense. For example they could write “the climate is temperate” (184) and “vegetation is typified” (186), for example.If the authors wish to clarify that they are only describing Stirling as it was when the study was conducted, they could make that clear: “At the time of this study, Stirling was…”

Lines 206-207: It would be helpful for the authors to define the exact time frame covered by the terms “early successional” and “late successional” since this can vary across forest types and geographic regions.

Line 234-242: Because it might not be clear to the generalist reader what the differences are between the transmitter types, the authors should describe why they used different transmission types and if they believe the collars were functionally equivalent.

Line 252: Unless I missed it, Lewis et al 2012 doesn’t seem to address acclimation.

Lines 292-298: The authors should check for range residency by fishers, include only range-resident fishers in their data set, and report that this check was performed. This can be done by examining the semivariogram of an animal’s movement track to see whether it reaches an asymptote (one way to do this is described here: https://cran.r-project.org/web/packages/ctmm/vignettes/variogram.html).

Lines 293, 295: I was unable to locate “Powell and Mitchell 2012” (which I’m assuming is in reference to the paper entitled “What is a Home Range?”) in the reference section.

Lines 304-307: While this may have been true historically, current best methods for estimating kernel bandwidth (i.e., Fleming et al. 2015 Ecology) explicitly take the first two of these factors into account by selecting a movement model appropriate for the species’ biology and accounting for telemetry error in the data. It is not clear to me why management goals should influence kernel bandwidth selection. I would recommend deleting this sentence, as it is not true and is also not necessary for justifying the author’s rationale for kernel bandwidth selection.

Line 240: The authors should define “isopleth”.

Line 453: The authors assume here that the number of males doesn’t limit reproduction, but Lewis et al 2012 concluded that it has a meaningful impact on reintroduction success. Is the assumption here in line with those findings? Was the number of males on the landscape higher than the threshold found by Lewis? The authors should clarify this (potential) discrepancy with past work.

Lines 517-521: I am confused as to why these 9 fishers are highlighted more than the other founder fishers. The authors should clarify or delete this information.

Line 536: Facka and Powell 2021 (in press) sometimes includes the phrase “in press” in the text, and sometimes does not.

Line 552: Line 551 just stated that there was no difference in habitat selection between the sexes. Were only females selecting habitat of higher value than available? Or were females the only sex for which habitat selection data was available? This statement needs clarification.

Line 590: The authors could describe how these fishers were chosen to be tested for anticoagulant rodenticides. If they were selected randomly that implies something different than if they were chosen because the researchers had reason to suspect rodenticide as a cause of death.

Line 607: The authors should clarify why they used 0.98 when it is higher than both estimates explained earlier in the paragraph.

Line 647: It’s troubling that extinction rates were so high for a juvenile survival rate of 0.2, since that is the rate taken from the literature. If the authors have any idea why their juvenile survival rate differs so drastically from the rate taken from the literature, and whether they have good reason to expect it to stay so high, it would be good to describe that here.

Line 675: Did the authors mean “...and where a newly established, functioning population was documented.”? If so, they should correct the sentence.

Discussion: Discussing each hypothesis in order would be more intuitive, rather than jumping around among them as the authors do now. If the authors have strongly held opinions that the discussion should be presented in this order, then the hypotheses can be re-ordered.

Lines 683-686: This analysis is based on a sample size of 1 fisher population. Given the weak evidence for this hypothesis and that it is more or less a minor digression from the main points of the manuscript, I would recommend that the authors delete this paragraph and the other text related to this hypothesis.

Line 742: There is no Figure 16.

Line 757: “Although…2017.” is a sentence fragment. The authors could cut “although” or add another clause making a contrasting point.

Line 765: “on” should be “of”

Line 829: If there are plans by California Fish and Wildlife to continue monitoring the population, it would be helpful for the authors to provide that information.

Line 873: The assertion that fishers can persist in urban environments is supported by the cited study (LaPoint et al 2013), but not supported by this paper’s findings; more careful wording here could clarify that (“This persistence is consistent with studies conducted on other fisher populations, one of which (LaPoint et al. 2013) suggested that fishers appear able to maintain populations on fragmented or patchy landscapes and even urban environments.”)

Table 7: Great presentation of information, but ordering the tree species from most utilized at the top to least utilized at the bottom could improve readability. Also, Quercus is spelled wrong multiple times.

Table 8: It is unclear to me how the model weights could sum to 0.8. Clarification would be helpful.

Figure 2: This figure could be improved by changing the legend so “historical range” is gray (the color of the map background) bordered by a black line rather than white bordered by a black line. Or the authors can make the line dashed or something else to indicate that it is the line that designates the historical range, not the white inside the line.

Figure 3: Overall good figure, but I am left with a couple questions. Why the dip in “medium tree open canopy” in ~2012? How did Sierra Pacific Industries calculate their projections? These are non-essential questions to answer, but it might be good to provide more information in the figure legend.

Figure 4: This has the potential to be an excellent figure, but minor adjustments would improve it. Switching the lines for 90% and 75% would be more intuitive (thus a kind of gradient would emerge, with 50% being the most solid line and 90% the least). The inset map of all of California in the bottom right corner would look better with a rectangle showing where the blown up portion of the map came from, rather than the rounded shape currently showing the location of Stirling.

Figure 5: This has the potential to be an excellent figure, but minor adjustments would improve it. The figure caption is good; in addition, an in-figure key with the different dashes and their meanings would improve the digestibility of the figure.

Figure 9: This figure could use some clarification in the caption. The meaning of “the proportions of home ranges of females and males that were logged in the preceding 10 years” is not clear until you read the rest of the results section. Why are those proportions divided by sex? Did we expect to see a difference between the proportions of ranges of males and females logged? What do the small numbers under each point of the black line represent? The authors could clarify the figure and caption so the reader isn’t left with these questions.

Figure 10: Overall good figure I do wonder how age distribution has changed over the years; the figure could be even more effective if this was demonstrated (Maybe with different colored bars for each year or a panel plot with a graph for each year).

Reviewer #2: This research and resulting manuscript comprise a significant contribution to our knowledge of the utility of and considerations during rare species' reintroductions and has vast management implications for current and future conservation actions for fishers and other rare carnivores. It represents an admirable undertaking towards a critical conservation tool that has lacked in evaluation over the years. Thus, this study fills major data gaps in our understanding of what factors effect the success of reintroductions. The methods and analyses used to arrive at the conclusions made are sound and thorough. Mainly, I have concerns about the structure and presentation of different points of the manuscript and composition of some of the ways data were interpreted. Below are specific recommendations:

Overall, the Introduction paints a good picture of the background that led to the reintroduction. I'm not sure that Lines 120-137 are completely necessary to the content and intent of the manuscript (yes, they make a case for Sierra Pacific, but also highlights the political and regulatory factors that influenced the reintroduction, which the authors may want to avoid), so if needed to shorten the Introduction, consider removing or reducing this section to the totally relevant details.

Line 66-67: This justification is not sound. The authors imply that poor habitat in a reintroduction site can be compensated for by other factors that have nothing to do with the potential for success of a reintroduction. Many arguments can be made that despite the research value provided by reintroduction attempts, a reintroduction that has poor potential from the beginning due to poor habitat is ill-advised and ethically a poor decision for animal welfare reasons, misuse of public funding, and misleading consequences for future conservation efforts of the species.

Paragraph starting with Line 69: given that you refer to "Establishment Phase" and then refer to that phase as well as the Persistence Phase in a figure, it makes sense to refer to and describe the "Persistence Phase" here so that the reader already understands the concept when they get to the Figure.

Line 83: change the second "are" to "and"

Lines 94 - 97: strange to switch to past tense when citing other literature, everything preceding and following that sentence is in present tense.

Lines 128 - 130: this sentence seems irrelevant; the mood and sentiments of different stakeholders doesn't seem to belong here, or in this manuscript at all.

Lines 158 - 159: This isn't an appropriate topic sentence for this paragraph where you are presenting all the hypotheses, as it's only related to one of the hypotheses. Either remove the sentence, or embed it after you present the hypothesis on site fidelity.

Lines 160 - 170: It is really important that the authors point out which hypotheses are being tested in this manuscript/ study, and which hypotheses have already been tested in previous literature. Yes, those hypotheses are cited appropriately. However, the reader is led to expect to see studies and analyses for each of these hypotheses in this manuscript. So it is disconcerting when that doesn't happen. I suggest clearly indicating that those were the 6 hypotheses for the overall study on the reintroduction and from the initiation of the reintroduction; then clearly say that only 3 of them are tested in this study/ manuscript. You can also indicate that the other 3 already-tested hypotheses will be reviewed in detail in this manuscript.

Methods:

Generally, since the methods are fairly complex and comprehensive, it would be ideal if the authors could somehow link each Hypothesis, as stated in the Table 1, to the methods described in each section. Perhaps after or right before presenting a new independent analysis, state which hypotheses are being addressed by that analysis.

Lines 207 - 209: should appear earlier in the Methods/ Study Area section, to show the landscape context of the study area.

Line 222: Gabriel et al 2012 doesn't seem to be an appropriate citation for this statement. It did not involve any disease exposure analysis. You should cite Gabriel et al. 2015, which you cite for a different reason later in the manuscript. *** Also, note that Gabriel et al. 2015 was not included in the Literature Cited section.

Lines 260 -261: Why did you vaccinate founders and Stirling-born fishers differently? Could that be a confounding factor in survival? Add an explanation here.

Lines 398 - 401: This hypothesis appears to represent a 7th independent hypothesis but isn't covered in the Introduction or Table where the hypotheses are presented.

Lines 480 - 490: Is this a standard method for calculating K? If so, please cite. If not, and since it's so critical to this study in particular, somehow justify your methods for K calculation.

Results:

Line 523: Do you mean you released the juveniles at their locations of capture or you moved them to the reintroduction site? If the latter, why? This is in opposition to the criteria for reintroduction stated earlier.

Lines 526-527: Numbers don't add up somehow. You identified the 133 as fishers born on Stirling. But the number of founders released at Stirling (presumably 40 based on the statement "thus, collected data on 173 fishers), is never identified. In fact, in the first sentences of the Results section, only the 9 removed from the Klamath region are discussed. What about the other 31?

Lines 540 - 542: Do these findings relate to the Thomasma indices of habitat? If so, the actual numerical indices should be cited here, then can be followed up with what those indices actually represented (closed canopy, small vs. large diameter, etc.)

Lines 552- 553: Though the figure you refer to shows this, it would be helpful if you mention the findings for males, right after you highlight that the females selected higher value habitat.

Line 560: Recap what each of these two population estimates come from (i.e. which methods of pop estimate)

Lines 560 - 563: Why state your predications of what you might have found in 2017 if you have no evidence to do so? Suggest you remove that statement.

Lines 566 - 567: "Our research effort..." does not seem necessary to point out here.

Line 575: "Also a few female...." State the exact number.

Line 606: missing a parentheses

Discussion:

Again, so much to discuss in relation to Hypotheses, can this be organized a bit to better address interpretations of the data and conclusions specific to the major hypotheses in the paper?

Line 683 - 686: This is more a general criticism: it really seems odd to state the conclusions to your predictions here, as if they were analyzed in this manuscript. Again, you should reorganize the hypotheses and address them specifically as hypotheses you addressed in this paper vs. those that were addressed in previous papers/ studies.

Line 690: Add Gabriel et al. 2015 for a more in-depth analysis of causes of mortality.

Line 694 and after: Prey availability was not evaluated so it's not appropriate to make a statement about prey availability not limiting survival. You can't be sure that prey availability had anything to do with survival or not, if you haven't measured it. Could be disconnect between prey and habitat, especially in ecological trap scenario.

Line 709: Did you mean to cite Figures 10, 12, 14 instead? figure 9 is irrelevant to the statement.

Line 717: Gabriel et al 2015 not in Lit Cited.

Line 714 - 726: I really don't think this paragraph is necessary or even relevant. Since no analysis on AR use or regulations were considered in any of the analyses, a deep dive into ARs is out of place.

Line 742: There was no Figure 16 in the manuscript.

Line 757-758: Not a full sentence, so is missing what seems to be a main point of that opening sentence.

Line 765: "on" should be "of"

Lines 793 - 801: Combine into one paragraph

Line 800 - 801: Appears to be mis-written. "...that changes in survival would have greater effects on survival..."

Line 840: Though the authors touch on the scarcity of porcupines in California, I don't think porcupines should even be brought up as important for fisher diet (like western gray squirrels are) since they are not a significant component of fisher diet today. We are not even sure they have ever been important for fisher in California, even though they are elsewhere.

Lines 863 - 865: These two sentences seem out of place in the context of what was being discussed before and after. Exclude, or move somewhere where it fits the context better.

Lines 866-867: Actually, the research did not do this. Because Sierra Pacific went above and beyond California Forest Practice Rules (CFPR), you will never know if a forested landscape sticking strictly to CFPR would have supported a new fisher population or not.

Line 881: In addition to others, more appropriate to cite Wengert 2013 (PhD dissertation focusing on predation in relation to forest management) rather than Wengert et al 2014.

Overall thoughts: you've convinced the reader that the fisher population, as derived from 40 founding fishers, has persisted on Sierra Pacific and has been successful into the Establishment Phase. However, if surrounding National Forest lands have not been monitored, is there a chance many animals (recruits) that you are not tracking have dispersed to nearby NF lands or other ownerships? consider adding some language about the unknowns and uncertainties - i.e. if given a choice of areas to settle, what if dispersing fishers actually choose NF lands over Sierra Pacific? You don't have the data to evaluate that, but you may add some sentences to the Discussion about how the fact that fishers were really only monitored on SPI lands isn't capturing the full nature of the reintroduction's success. In truth, it may also be more successful than stated, if it is a continuing source population for adjacent lands. Or less; it just can't be known, but state that.

Table 1. I already strongly suggested indicating in the body of the ms which hypotheses were tested in this paper and which were already tested and published. I think you should do that here in this Table as well.

Hypothesis 3: This is contingent on knowing what constitutes an "independent population." That should be discussed somewhere (probably not in the Table) - perhaps in the Introduction.

table 6. End was cut off; ensure a complete Table is provided.

Figure 10. Is this typical of demographic distribution of a fisher population? Or does this indicate a population with much dispersal away from the study area (thus, animals that stay within the study area are less likely to make it to adulthood)? Please touch on the significance of this distribution a bit more. The graph tends to make it look like a harvested population, though we know it's not.

Figure 9. What do the small numbers below the top line represent? Sample sizes of fishers? The wording is also very confusing: “utilization distributions with > 50% within Stirling.” Can you explain what that means exactly?

6. PLOS authors have the option to publish the peer review history of their article (what does this mean? ). If published, this will include your full peer review and any attached files.

**Do you want your identity to be public for this peer review?** For information about this choice, including consent withdrawal, please see our Privacy Policy .

Reviewer #1: No

Reviewer #2: No

---

## [Author Response · Author response to Decision Letter 1]

2 Sep 2024

Reviewer #1: Summary:

This manuscript is a sort of capstone monograph describing the lessons learned from the final decade of the lead authors’ career before transitioning to emeritus (where he has continued to be productive in retirement). In this manuscript, the authors document the reintroduction of fishers into land in California that is managed for timber. Monitoring the reintroduced population for 8 years showed promising monthly survival and reproduction rates. The population remained small (<70) but grew steadily throughout the study. The authors performed simulation modeling and predicted that the population is unlikely to go extinct over the next 40 years. This study is a very thorough consideration of the feasibility of reintroducing fishers into less-than-ideal habitat, as well as a broader exploration of the value of using reintroductions to assess resource use and population dynamics of a species. The recommendation that “non ideal” habitat be considered for fisher reintroductions is well supported. There is also strong support for the authors’ argument that re-establishing a population should not be the sole goal of a reintroduction. This manuscript covered complex concepts in mostly excellent, clear prose. This is a solid, thorough manuscript–my quibbles are mostly minor and are listed below.

Minor comments:

Line 42: I think the authors mean "wildfire" instead of "wildlife" here. If so, they should correct the text to prevent potential confusion.

Changed.

Line 67: The argument that ease of access or detailed habitat information compensates for lower habitat quality needs to be qualified. This can contribute to the ability of the habitat to make a great study site, but it does not contribute to the ability of the habitat to support the species.

We agree with the reviewer and have revised the sentence to discuss biologically relevant considerations that might offset poor quality habitat for consideration.

Line 83: “and are” rather than “are”.

Changed.

Lines 91-93: Some kinds of grouse (e.g., ruffed) and snowshoe hares prefer, or at least do fine in, early-successional forest. I do not think it is accurate to say that these are mature-forest species, so I would recommend deleting this sentence.

We have revised this paragraph to reflect that these species are often found in areas that overlap fisher ranges but are not exclusively found in such areas.

Line 94: The in text citation “Buskirk and Powell 1994” seems to refer to the reference in line 1169 (a chapter authored by Powell from a volume written by Buskirk, Harestad, Raphael, and Powell). If so, the in-text citation should read “Powell 1994.”

Buskirk and Powell 1994 has been added to Lit Cited.

Line 115: It is unclear what “close genetic relatedness” means in this context. Some clarification would be helpful.

We have changed this sentence to state that the analyses only used distance between release and original populations as indices of genetic relatedness.

Line 120-137: These paragraphs contain background that is arguably not necessary and could be cut or severely reduced. If the authors feel strongly that it is necessary information, it would probably be better suited for the Materials and Methods section, under the “study site” subheading.

We have changed this paragraph to reduce much of the background, cited a dissertation which includes much of the underlying conditions, and focused on the information we think is salient to the paper.

Lines 158-172: The rationale for these hypotheses is not always clear. They would benefit from some extra detail in a “Because ABC, then we hypothesized XYZ” construction.

We have revised this paragraph per the reviewer’s comment but have not gone into extreme detail to try and keep the introduction relatively short.

Line 165: It is unclear what “moved before 1 January” means. The authors could perhaps clarify by saying “translocated before 1 January of a respective year.”

We have explained here the 3 hypotheses that have been handled elsewhere and cited those papers. The 1 January date is now explained.

Line 173: The authors should consider adding the effects of fire as a 7th hypothesis; the current paragraph is underdeveloped and feels like an afterthought.

We have added more information on our rationale for including an analysis on wildfire but could not test any hypotheses generated by our simulations. We note elsewhere the hypotheses stimulated by those simulations.

Line 180: “Field Methods” implies another Methods section, perhaps titled “Analytical Methods”. No such section appears to exist. Therefore I recommend changing the title at line 180 to “Materials and Methods” or “Methods” and then using subheadings, if desired, for field and analytical methods.

Changed to just “Methods”, as this section also has analysis methods

Lines 181-209: Even though the authors use past tense throughout the paper, in this section they are describing an existing landscape. It would make more sense to the reader if they were to describe it in present tense. For example they could write “the climate is temperate” (184) and “vegetation is typified” (186), for example.If the authors wish to clarify that they are only describing Stirling as it was when the study was conducted, they could make that clear: “At the time of this study, Stirling was…”

We have added the phrase “during our study” to suggest we are only describing conditions that occurred during this period. While we agree that some things are likely to be unchanging for the immediate future, many things like the amounts of timber, ownership, etc could potentially change, even before this manuscript is published.

Lines 206-207: It would be helpful for the authors to define the exact time frame covered by the terms “early successional” and “late successional” since this can vary across forest types and geographic regions.

We fixed a typographical error and defined successional stages.

Line 234-242: Because it might not be clear to the generalist reader what the differences are between the transmitter types, the authors should describe why they used different transmission types and if they believe the collars were functionally equivalent.

We have added our rationale behind choosing the transmitters.

Line 252: Unless I missed it, Lewis et al 2012 doesn’t seem to address acclimation.

Lewis et al. tested hard vs soft releases. We have slightly revised this section to make that explicit.

Lines 292-298: The authors should check for range residency by fishers, include only range-resident fishers in their data set, and report that this check was performed. This can be done by examining the semivariogram of an animal’s movement track to see whether it reaches an asymptote (one way to do this is described here: https://cran.r-project.org/web/packages/ctmm/vignettes/variogram.html).

We effectively did this analyses (Facka and Powell 2021). Both male and females (reintroduced and Stirling born) animals settled into home ranges within about 500 days after capture or release from trapping.

Lines 293, 295: I was unable to locate “Powell and Mitchell 2012” (which I’m assuming is in reference to the paper entitled “What is a Home Range?”) in the reference section.

Added.

Lines 304-307: While this may have been true historically, current best methods for estimating kernel bandwidth (i.e., Fleming et al. 2015 Ecology) explicitly take the first two of these factors into account by selecting a movement model appropriate for the species’ biology and accounting for telemetry error in the data. It is not clear to me why management goals should influence kernel bandwidth selection. I would recommend deleting this sentence, as it is not true and is also not necessary for justifying the author’s rationale for kernel bandwidth selection.

Revised.

Line 240: The authors should define “isopleth”.

We have modified this sentence to be more precise.

Line 453: The authors assume here that the number of males doesn’t limit reproduction, but Lewis et al 2012 concluded that it has a meaningful impact on reintroduction success. Is the assumption here in line with those findings? Was the number of males on the landscape higher than the threshold found by Lewis? The authors should clarify this (potential) discrepancy with past work.

All males except 1 released were adult males and, therefore, we could assume that males were not limiting.

Lines 517-521: I am confused as to why these 9 fishers are highlighted more than the other founder fishers. The authors should clarify or delete this information.

Revised

Line 536: Facka and Powell 2021 (in press) sometimes includes the phrase “in press” in the text, and sometimes does not.

These are two papers. The “in press” paper has been published and is now cited as such.

Line 552: Line 551 just stated that there was no difference in habitat selection between the sexes. Were only females selecting habitat of higher value than available? Or were females the only sex for which habitat selection data was available? This statement needs clarification.

We revised to give information for males.

Line 590: The authors could describe how these fishers were chosen to be tested for anticoagulant rodenticides. If they were selected randomly that implies something different than if they were chosen because the researchers had reason to suspect rodenticide as a cause of death.

We revised the Methods section to state that all fishers in appropriate condition were tested.

Line 607: The authors should clarify why they used 0.98 when it is higher than both estimates explained earlier in the paragraph.

We revised this sentence to make clear how 0.98 was used.

Line 647: It’s troubling that extinction rates were so high for a juvenile survival rate of 0.2, since that is the rate taken from the literature. If the authors have any idea why their juvenile survival rate differs so drastically from the rate taken from the literature, and whether they have good reason to expect it to stay so high, it would be good to describe that here.

We have an hypotheses for this that we present in Discussion. Briefly, female mustelids with long delayed implantation conceive when the environmental conditions next year are unknown. From conception through birth, females trim litter size to fit environmental conditions. If fecundity is measured by counting corpora lutea, litter size at birth can be greatly over estimated.

Line 675: Did the authors mean “...and where a newly established, functioning population was documented.”? If so, they should correct the sentence.

Revised.

Discussion: Discussing each hypothesis in order would be more intuitive, rather than jumping around among them as the authors do now. If the authors have strongly held opinions that the discussion should be presented in this order, then the hypotheses can be re-ordered.

We have revised the entire manuscript to deal with hypotheses in their order and not to confuse readers with tests of hypotheses already published.

Lines 683-686: This analysis is based on a sample size of 1 fisher population. Given the weak evidence for this hypothesis and that it is more or less a minor digression from the main points of the manuscript, I would recommend that the authors delete this paragraph and the other text related to this hypothesis.

Deleted as part of the revisions to get hypotheses in order.

Line 742: There is no Figure 16.

Figures and tables have been renumbered due to revising. This problem has been fixed.

Line 757: “Although…2017.” is a sentence fragment. The authors could cut “although” or add another clause making a contrasting point.

Revised.

Line 765: “on” should be “of”’

Revised.

Line 829: If there are plans by California Fish and Wildlife to continue monitoring the population, it would be helpful for the authors to provide that information.

No such plans to our knowledge, though we would certainly like to go back and pick up the research. Recent fires have entered Stirling: the Camp fire in 2018 and the fire raging there as I type (27 July 2024). We would like to learn how our predictions about fire hold up.

Line 873: The assertion that fishers can persist in urban environments is supported by the cited study (LaPoint et al 2013), but not supported by this paper’s findings; more careful wording here could clarify that (“This persistence is consistent with studies conducted on other fisher populations, one of which (LaPoint et al. 2013) suggested that fishers appear able to maintain populations on fragmented or patchy landscapes and even urban environments.”)

Revised.

Table 7: Great presentation of information, but ordering the tree species from most utilized at the top to least utilized at the bottom could improve readability. Also, Quercus is spelled wrong multiple times.

Fixed spelling, sorted.

Table 8: It is unclear to me how the model weights could sum to 0.8. Clarification would be helpful.

Rounding.

Figure 2: This figure could be improved by changing the legend so “historical range” is gray (the color of the map background) bordered by a black line rather than white bordered by a black line. Or the authors can make the line dashed or something else to indicate that it is the line that designates the historical range, not the white inside the line.

Revised.

Figure 3: Overall good figure, but I am left with a couple questions. Why the dip in “medium tree open canopy” in ~2012? How did Sierra Pacific Industries calculate their projections? These are non-essential questions to answer, but it might be good to provide more information in the figure legend.

Revised

Figure 4: This has the potential to be an excellent figure, but minor adjustments would improve it. Switching the lines for 90% and 75% would be more intuitive (thus a kind of gradient would emerge, with 50% being the most solid line and 90% the least). The inset map of all of California in the bottom right corner would look better with a rectangle showing where the blown up portion of the map came from, rather than the rounded shape currently showing the location of Stirling.

Revised

Figure 5: This has the potential to be an excellent figure, but minor adjustments would improve it. The figure caption is good; in addition, an in-figure key with the different dashes and their meanings would improve the digestibility of the figure.

The legend has the dashes.

Figure 9: This figure could use some clarification in the caption. The meaning of “the proportions of home ranges of females and males that were logged in the preceding 10 years” is not clear until you read the rest of the results section. Why are those proportions divided by sex? Did we expect to see a difference between the proportions of ranges of males and females logged? What do the small numbers under each point of the black line represent? The authors could clarify the figure and caption so the reader isn’t left with these questions.

Figure and legend revised.

Figure 10: Overall good figure I do wonder how age distribution has changed over the years; the figure could be even more effective if this was demonstrated (Maybe with different colored bars for each year or a panel plot with a graph for each year).

Revised

Reviewer #2:

This research and resulting manuscript comprise a significant contribution to our knowledge of the utility of and considerations during rare species' reintroductions and has vast management implications for current and future conservation actions for fishers and other rare carnivores. It represents an admirable undertaking towards a critical conservation tool that has lacked in evaluation over the years. Thus, this study fills major data gaps in our understanding of what factors effect the success of reintroductions. The methods and analyses used to arrive at the conclusions made are sound and thorough. Mainly, I have concerns about the structure and presentation of different points of the manuscript and composition of some of the ways data were interpreted. Below are specific recommendations:

Overall, the Introduction paints a good picture of the background that led to the reintroduction. I'm not sure that Lines 120-137 a

---

## [Decision Letter · Decision Letter 1]

4 Oct 2024

PONE-D-24-04811R1Establishing a Carnivoran of Extensive Forests on an Intensively Managed Landscape: 1

Habitat and Population EstablishmentPLOS ONE

Dear Dr. Powell,

Thank you for submitting your manuscript to PLOS ONE. After careful consideration, we feel that it has merit but does not fully meet PLOS ONE’s publication criteria as it currently stands. Therefore, we invite you to submit a revised version of the manuscript that addresses the points raised during the review process.

**I appreciate the authors’ thoroughness and thoughtfulness in addressing the numerous comments and suggestions by the original reviewers. This paper evaluates the potential influence of habitat quality on the reintroduction of fishers, a carnivore of conservation concern.  While it makes an important and timely contribution to our understanding of how to improve reintroduction efforts in managed lands the manuscript needed a little more clarification, some streamlining, potential reorganization of the sections of the paper, and some additional context.  The revised version is much clearer and the results are much better supported.  The improved flow and refined attention to details is notable.  As the reviewer suggests there are few minor revisions that could further strengthen the paper and improve the readability.  Please note the additional information needed on data availability.  With these minor revisions, the paper should be well crafted to contribute to our understanding of the factors that facilitate successful reintroduction of fishers.**

We look forward to receiving your revised manuscript.

Kind regards,

Karen Root, Ph.D.

Academic Editor

PLOS ONE

**Journal Requirements:**

Reviewers' comments:

Reviewer's Responses to Questions

**Comments to the Author**

1. If the authors have adequately addressed your comments raised in a previous round of review and you feel that this manuscript is now acceptable for publication, you may indicate that here to bypass the “Comments to the Author” section, enter your conflict of interest statement in the “Confidential to Editor” section, and submit your "Accept" recommendation.

Reviewer #1: (No Response)

2. Is the manuscript technically sound, and do the data support the conclusions?

Reviewer #1: Yes

3. Has the statistical analysis been performed appropriately and rigorously? 

Reviewer #1: Yes

4. Have the authors made all data underlying the findings in their manuscript fully available?

Reviewer #1: No

5. Is the manuscript presented in an intelligible fashion and written in standard English?

Reviewer #1: Yes

6. Review Comments to the Author

**Reviewer #1: ** Summary:

In this manuscript, the authors document the reintroduction of fishers into land in California that is managed for timber. Monitoring the reintroduced population for 8 years showed promising monthly survival and reproduction rates. The population remained small (<70) but grew steadily throughout the study. The authors performed simulation modeling and predicted that the population is unlikely to go extinct over the next 40 years. This study is a very thorough consideration of the feasibility of reintroducing fishers into less-than-ideal habitat, as well as a broader exploration of the value of using reintroductions to assess resource use and population dynamics of a species. The recommendation that “non ideal” habitat be considered for fisher reintroductions is well supported. There is also strong support for the authors’ argument that re-establishing a population should not be the sole goal of a reintroduction. The science of this manuscript is solid and generally well-explained. The authors responded well to the reviewer comments. They improved the organization of the manuscript by clarifying which hypotheses they tested and by providing and standardizing section headings. Below I have listed only a few minor recommendations that would increase the clarity and readability of this manuscript.

Minor comments:

Lines 153-167: Authors added a list of hypotheses and standardized their order throughout the paper, as requested by the reviewers. This significantly improved the flow and readability. However, the presentation of hypotheses in the introduction could be standardized for clarity. For example, Hypothesis 1 is presented as an “if…then” statement, while Hypothesis 3 is presented as a prediction.

Line 181: I think the authors mean “has” rather than “have”.

Line 298: “Thereafter, 2 times per year to locate fishers.” is a sentence fragment. The authors could replace it with a sentence such as: “Thereafter, we located fishers only two times per year.”

Line 641: I believe that this line should say “Hypothesis 4” rather than “Hypothesis 3”.

Figure 4: The figure description mentions that colored stars represent release sites, while in the figure release sites are represented by white shapes.

Data Availability: Writing simply that data is available from CDFW is not adequate for data availability. If the data are owned by CDFW, then the authors need to provide contact information and the process for getting access to the data.

7. PLOS authors have the option to publish the peer review history of their article (what does this mean? ). If published, this will include your full peer review and any attached files.

**Do you want your identity to be public for this peer review?** For information about this choice, including consent withdrawal, please see our Privacy Policy .

Reviewer #1: No

---

## [Author Response · Author response to Decision Letter 2]

14 Feb 2025

BELOW ARE THE AUTHORS’ RESPONSES TO THE REVIEWER’S COMMENTS

Reviewer #1: Summary:

In this manuscript, the authors document the reintroduction of fishers into land in California that is managed for timber. Monitoring the reintroduced population for 8 years showed promising monthly survival and reproduction rates. The population remained small (<70) but grew steadily throughout the study. The authors performed simulation modeling and predicted that the population is unlikely to go extinct over the next 40 years. This study is a very thorough consideration of the feasibility of reintroducing fishers into less-than-ideal habitat, as well as a broader exploration of the value of using reintroductions to assess resource use and population dynamics of a species. The recommendation that “non ideal” habitat be considered for fisher reintroductions is well supported. There is also strong support for the authors’ argument that re-establishing a population should not be the sole goal of a reintroduction. The science of this manuscript is solid and generally well-explained. The authors responded well to the reviewer comments. They improved the organization of the manuscript by clarifying which hypotheses they tested and by providing and standardizing section headings. Below I have listed only a few minor recommendations that would increase the clarity and readability of this manuscript.

Minor comments:

Lines 153-167: Authors added a list of hypotheses and standardized their order throughout the paper, as requested by the reviewers. This significantly improved the flow and readability. However, the presentation of hypotheses in the introduction could be standardized for clarity. For example, Hypothesis 1 is presented as an “if…then” statement, while Hypothesis 3 is presented as a prediction.

Line 181: I think the authors mean “has” rather than “have”.

- Changed

Line 298: “Thereafter, 2 times per year to locate fishers.” is a sentence fragment. The authors could replace it with a sentence such as: “Thereafter, we located fishers only two times per year.”

- Changed

Line 641: I believe that this line should say “Hypothesis 4” rather than “Hypothesis 3”.

- Changed

Figure 4: The figure description mentions that colored stars represent release sites, while in the figure release sites are represented by white shapes.

- Changed

Data Availability: Writing simply that data is available from CDFW is not adequate for data availability. If the data are owned by CDFW, then the authors need to provide contact information and the process for getting access to the data.

- Data have been uploaded to MoveBank

---

## [Editor Report · Decision Letter 2]

21 Feb 2025

Establishing a Late Successional Carnivoran on an Intensively Managed Landscape:

Habitat and Population Establishment

PONE-D-24-04811R2

Dear Dr. Powell,

We’re pleased to inform you that your manuscript has been judged scientifically suitable for publication and will be formally accepted for publication once it meets all outstanding technical requirements.

Kind regards,

Karen Root, Ph.D.

Academic Editor

PLOS ONE

Additional Editor Comments (optional):

I appreciate the authors’ thoroughness and thoughtfulness in addressing the numerous comments and suggestions by the reviewers! With these revisions the paper is now suitable for publication.
---

## [Editor Report · Acceptance letter]

PONE-D-24-04811R2

PLOS ONE

Dear Dr. Powell,

I'm pleased to inform you that your manuscript has been deemed suitable for publication in PLOS ONE. Congratulations! Your manuscript is now being handed over to our production team.

Kind regards,

on behalf of

Professor Karen Root

Academic Editor

PLOS ONE